# NEURAL NETWORKS AND THE CHOMSKY HIERARCHY

**Grégoire Delétang**[*,1] **Anian Ruoss**[*,1] **Jordi Grau-Moya**[1] **Tim Genewein**[1] **Li Kevin Wenliang**[1]

**Elliot Catt**[1] **Chris Cundy**[†,2] **Marcus Hutter**[1] **Shane Legg**[1] **Joel Veness**[1] **Pedro A. Ortega**[†]

## ABSTRACT

Reliable generalization lies at the heart of safe ML and AI. However, understanding when and how neural networks generalize remains one of the most important unsolved problems in the field. In this work, we conduct an extensive empirical study (20 910 models, 15 tasks) to investigate whether insights from the theory of computation can predict the limits of neural network generalization in practice. We demonstrate that grouping tasks according to the Chomsky hierarchy allows us to forecast whether certain architectures will be able to generalize to out-of-distribution inputs. This includes negative results where even extensive amounts of data and training time never lead to any non-trivial generalization, despite models having sufficient capacity to fit the training data perfectly. Our results show that, for our subset of tasks, RNNs and Transformers fail to generalize on non-regular tasks, LSTMs can solve regular and counter-language tasks, and only networks augmented with structured memory (such as a stack or memory tape) can successfully generalize on context-free and context-sensitive tasks.

## 1 INTRODUCTION

Statistical learning theory is the most widely used theory of generalization in practical machine learning, justifying empirical risk minimization and estimating the generalization error via a test set (Vapnik, 1998). However, its central assumption that training and test data are independent and identically distributed (i.i.d.) is violated for many problems of interest (distribution shifts, continual learning, etc.). An example of such a non-i.i.d. setting is testing generalization on sequence prediction problems, where an agent is trained with sequences of length $\ell \leq N$ and tested with arbitrarily longer sequences $\ell \gg N$. This problem is of particular importance since it subsumes all computable problems (Dawid, 1984; Rich, 2007; Sipser, 1997; Solomonoff, 2009; 2010). Central to sequence prediction is *inductive inference*, which consists of deriving a general rule from a finite set of concrete instances and using this rule to make predictions. For example, in *program induction* (Goldberg, 1989; Gomez et al., 2008; Holland, 1992; Liang et al., 2013; Nordin, 1997; Solomonoff, 1964a;b; Wineberg & Oppacher, 1994), the goal is to obtain a model that correctly identifies the underlying data-generating process given examples of input-output sequences. Then, if the model is correct, it can produce results in accordance with the generative process for previously unseen input sequences.

The key challenge of inductive inference (as opposed to deduction) is that it does not allow selecting one hypothesis with certainty among the ones that fit the data. For instance, the sequence $2, 4, 6, 8$ has infinitely many possible continuations. Thus, any principle that selects a particular continuation requires additional assumptions that are independent of the data, i.e., inductive biases (Mitchell, 1980). In machine learning, the network architecture, training mechanisms (e.g., gradient descent), and initial distributions over parameters all generate their corresponding inductive biases. This has led to a vast number of approaches for designing inductive biases via architectural and training protocol changes (see Battaglia et al. (2018) for an overview). However, the problem is that stronger inductive biases generally come at the cost of decreasing the universality of a model, and thus finding a good balance between the two is one of the biggest challenges in the contemporary literature.

Even if a neural architecture is theoretically universal or Turing complete, gradient-based training, which cannot exhaustively search the parameter space, can impede finding the right solution and thus

---

[*]Equal contribution. Correspondence to {gdelt, anianr}@deepmind.com.
[1]DeepMind. [2]Stanford University. [†]Work performed while the author was at DeepMind.

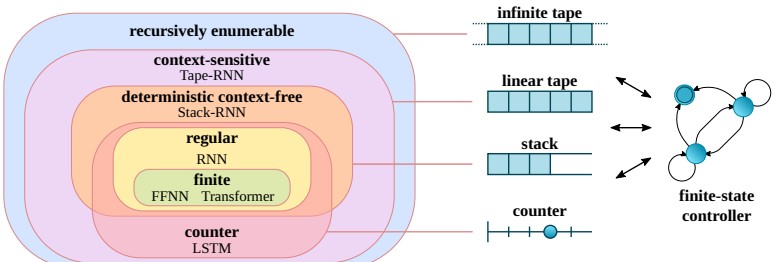

Figure 1: Formal language classes and their correspondence with neural network architectures. *Left*: Our empirical evaluation locates the architectures on the hierarchy of formal language classes. *Right*: Each formal language class is associated with a minimal computational model (automaton) to recognize or generate the language (see Section 3). All automata have a finite-state controller at their core, in addition to increasingly restrictive memory access as we descend the hierarchy.

practically render the model non-universal. Therefore, both architectural and training limitations impact which sequence prediction problems a model can solve in practice. In formal language theory, the Chomsky hierarchy (Chomsky, 1956) classifies such (sequence prediction) problems by increasing complexity. This hierarchy is associated with an equivalent hierarchy of models (automata) that can solve different problem classes (Savage, 1998; Sipser, 1997). Lower-level automata have restrictive memory models and can only solve lower-level problems, while Turing machines with infinite memory and unrestricted memory access lie on top of the hierarchy and can solve all computable problems. However, unlike for classical automata, a unified placement of neural architectures on the Chomsky hierarchy has not yet been practically established, which is precisely the goal of our work.

**This work** We conduct an extensive empirical study with the aim of discovering how neural network models used for program induction relate to the idealized computational models defined by the Chomsky hierarchy *in practice* (see Fig. 1 for a summary of our findings). We investigate whether the theoretical limitations of certain neural models hold in practice when trained with gradient-based methods. For example, previous work has theoretically argued that RNNs are Turing complete (Siegelmann & Sontag, 1994). However, more recent theoretical analyses (Ackerman & Cybenko, 2020; Merrill, 2019; Weiss et al., 2018) showed that RNNs lie much lower on the Chomsky hierarchy. To complement these theoretical analyses, we conduct a large-scale empirical evaluation on sequence prediction problems. We make the following main contributions:

- We conduct an extensive generalization study (20 910 models, 15 tasks) of state-of-the-art neural network architectures (RNN, LSTM, Transformer) and memory-augmented networks (Stack-RNN, Tape-RNN) on a battery of sequence-prediction tasks spanning the entire Chomsky hierarchy that can be practically tested with finite-time computation.

- We open-source a length generalization benchmark (`https://github.com/deepmind/neural_networks_chomsky_hierarchy`) that is out of reach for state-of-the-art sequence prediction models and allows us to pinpoint the failure modes of these architectures.

- We show how increasing amounts of training data do not enable generalization on our tasks higher up in the hierarchy for some architectures (under sufficient capacity to perfectly learn the training data) potentially implying hard limitations for scaling laws (Kaplan et al., 2020).

- We demonstrate how augmenting architectures with differentiable structured memory (e.g., with a stack or a tape) can enable them to solve tasks higher up the hierarchy.

## 2 RELATED WORK

**Learning formal languages** A long line of work has empirically investigated whether common machine learning architectures, including RNNs (Elman, 1990), GRUs (Cho et al., 2014), SCNs (Giles et al., 1992; Pollack, 1991), LSTMs (Hochreiter & Schmidhuber, 1997), and Transformers (Vaswani et al., 2017), are capable of learning formal languages. The main insights are: These networks can

learn simple counting languages (Hölldobler et al., 1997; Steijvers & Grünwald, 1996; Wiles & Elman, 1995; Rodriguez & Wiles, 1997) and the Dyck-1 language (Skachkova et al., 2018; Suzgun et al., 2019a). To learn more advanced languages, RNNs and LSTMs require exponential memory in terms of the input length (Sennhauser & Berwick, 2018), as they lack an external memory structure. They are capable of learning simple context-sensitive languages, such as $a^n b^n c^n$, but in a very limited way, i.e., they generalize only to lengths close to those seen during training (Bodén & Wiles, 2000; 2002; Gers & Schmidhuber, 2001). Similarly, Transformers cannot learn Dyck-$n$ languages for $n > 1$ and long sequences (Ebrahimi et al., 2020). Concurrent work also studies length generalization on synthetic reasoning tasks (akin to learning formal languages) but for pretrained large language models (Anil et al., 2022; Zhang et al., 2022). Thus, while prior work has investigated a single architecture on a restricted set of tasks under different experimental conditions, we provide a unified experimental protocol that spans all the levels of the Chomsky hierarchy for a wide range of models.

**Neural networks and the Chomsky hierarchy**   It was theoretically shown that RNNs and Transformers are Turing complete (Chen et al., 2018; Pérez et al., 2019; 2021; Siegelmann & Sontag, 1994). However, these results are impractical as they rely on an unbounded number of recurrent steps and on arbitrary numerical precision. Thus, more recent work (Ackerman & Cybenko, 2020; Bhattamishra et al., 2020; Hahn, 2020; Hao et al., 2022; Korsky & Berwick, 2019; Merrill, 2019; Merrill et al., 2020; Merrill & Sabharwal, 2022; Weiss et al., 2018) has refined these theoretical analyses by considering linear computation steps and logarithmic precision, showing that: (i) RNNs and GRUs can, in theory, recognize regular languages, and (ii) LSTMs are strictly more powerful since they can learn a counting mechanism (i.e., are k-counter machines). Moreover, it was theoretically shown that Transformers are not well-aligned with the Chomsky hierarchy since they cannot recognize certain regular languages (e.g., periodic finite-state languages), while being able to learn some counter languages (e.g., Shuffle-Dyck and $n$-ary Boolean expressions). A different approach proposed a computational model to capture the Transformer operations and used it to show which tasks could conceivably be learned by a Transformer (histograms, sorting, Dyck languages) (Weiss et al., 2021). However, this approach only upper-bounds the capabilities of a model and does not provide any insight on whether gradient-based methods will find parameters that can solve a task in practice, which is precisely the goal of our work. In that sense, our work complements the above studies by investigating how well gradient-based learning can exploit the inductive biases of common machine learning architectures to recognize languages on different levels of the Chomsky hierarchy.

**Memory-augmented networks**   A popular approach for augmenting neural architectures with external memory is to equip a recurrent network with read and write access to differentiable memory structures, such as deterministic stacks (Das et al., 1992a;b; Grefenstette et al., 2015; Hao et al., 2018; Joulin & Mikolov, 2015; Mali et al., 2021; Mozer & Das, 1992; Stogin et al., 2020; Sun et al., 1993; Suzgun et al., 2019b; Yogatama et al., 2018), nondeterministic stacks (DuSell & Chiang, 2020; 2022), random access memory (Danihelka et al., 2016; Kurach et al., 2016), and memory matrices (Graves et al., 2014; 2016; Gülçehre et al., 2018; Yang & Rush, 2017). Such memory-augmented networks are capable of recognizing complex languages, including tasks like copying or reversing strings, but they can only perform one computation step per input token. Thus, to solve superlinear tasks (e.g., binary multiplication), memory-augmented networks also need to be able to perform a linear number of steps per token (Freivalds & Liepins, 2017; Kaiser & Sutskever, 2016; Price et al., 2016). In a different effort, reinforcement learning has been applied to interact with discrete memory interfaces, making deterministic memory operations amenable to gradient-based training (Zaremba & Sutskever, 2015; Zaremba et al., 2016). Finally, prior work also proposed networks with read-only memory access (Sukhbaatar et al., 2015; Weston et al., 2015) or spatiotemporal connections (Kalchbrenner et al., 2016). We also consider memory-augmented networks and find that memory structures are necessary to generalize on sequence prediction problems higher up the Chomsky hierarchy.

## 3   BACKGROUND

We now provide the necessary background on how to link sequence prediction and formal language theory, allowing us to associate neural architectures and the Chomsky hierarchy. As we want to evaluate the capability of networks to generalize to sequences of unseen length, we need a way of generating arbitrarily long sequences that all exhibit the same structure. To that end, we will consider sequences as elements of infinite formal languages, which are generated by a grammar.

Table 1: Chomsky hierarchy grammar types, their corresponding minimal automata and memory structures required for language recognition or generation, their production rules, and the corresponding example languages from Section 3. The production rules symbols are defined as: $a$ terminal; $A, B$ non-terminal; $\alpha, \beta, \gamma$ string of terminals and/or non-terminals ($\alpha, \beta$ can be empty, but not $\gamma$).

| Grammar type (low → high) | Automaton | Memory | Production rules | Example |
|---|---|---|---|---|
| Regular (R) | Finite-state automaton (FSA) | Automaton state | $A \to a \mid aB$ | A |
| Context-free (CF) | Push-down automaton (PDA) | + infinite stack (only top entry accessible) | $A \to \alpha$ | B |
| Context-sensitive (CS) | Linear bounded automaton (LBA) | + bounded tape (all entries accessible) | $\alpha A \beta \to \alpha \gamma \beta$ | C |
| Recursively enumerable (RE) | Turing machine (TM) | + infinite tape (all entries accessible) | $\gamma \to \alpha$ | D |

**Formal language** A formal language $L$ is a set of words, where each word is a finite string of symbols drawn from a finite alphabet $\Sigma$. For example, with a binary alphabet $\Sigma = \{0, 1\}$ we can describe the following infinite languages: $A = \{0^n 1^m \mid n, m > 0\}$, $B = \{w \mid w$ is a palindrome$\}$, $C = \{ww \mid w$ is a word$\}$, and $D = \{w \mid w$ describes a terminating Turing machine$\}$.

**Generative grammar** Languages of infinite cardinality can be generated with a finite alphabet and a finite generative grammar. A formal grammar (Chomsky, 1956) is defined by a finite set of terminal symbols, a finite set of non-terminal symbols, a distinctive start symbol, and a finite set of production rules that map from non-terminal symbols to other symbols. The set of all words that can be formed by such a grammar is its corresponding formal language. For example, the binary palindrome language $B$ can be generated by the set of rules: $P \to \epsilon \mid 0 \mid 1 \mid 0P0 \mid 1P1$, with start and nonterminal symbol $P$ and terminal symbols $\{0, 1\}$. The Chomsky hierarchy arises from categorizing the allowed relationships between terminals and non-terminals in the grammar (see Table 1). For example, $B$ is not regular, since the right-hand side contains a variable enclosed by two terminal symbols, but context-free, since the left-hand side consists of a single nonterminal symbol.

**Recognizing a language** An automaton recognizes a language if it accepts all strings in the language and no others. For example, a (deterministic) finite-state automaton finite-state automaton (FSA) can be defined as a 5-tuple $(Q, \Sigma, \delta, q_0, F)$, consisting of a finite set of states $Q$, the finite alphabet $\Sigma$, a transition function $\delta : Q \times \Sigma \to Q$, an initial state $q_0 \in Q$, and a set of accept states $F \subseteq Q$. Thus, an FSA accepts a string $w = a_1 a_2 \ldots a_n$ over the alphabet $\Sigma$ if there exists a sequence of states $r_0, r_1, \ldots, r_n \in Q$ such that $r_0 = q_0$, $r_{i+1} = \delta(r_i, a_{i+1})$ for $i = 0, \ldots, n-1$, and $r_n \in F$. More complex automata, such as the push-down automaton (PDA), additionally have access to an external memory structure. The Chomsky hierarchy classifies languages based on the automaton that can recognize it (see Table 1). All automata in the hierarchy have a finite state controller at their core, with increasingly flexible memory structures. Language classes generated by higher grammar types subsume languages generated by those lower in the hierarchy, e.g., all regular languages are CF, all CF languages are CS, etc. For technical reasons, we differentiate deterministic context-free (DCF) and nondeterministic context-free (NDCF) languages, since nondeterministic push-down automata (PDAs) can recognize some CF languages that deterministic PDAs cannot. We also consider (real-time) counter languages, which are recognized by finite automata with one or more (infinite) counters (or, equivalently, a PDA with one or more stacks and a single stack symbol). Counter languages are a proper superset of regular languages and a proper subset of CS languages (see Fig. 1).

**Transduction vs. recognition** In practice, learning to recognize a formal language by training a classifier on words and non-words is hard because there is, in general, no formal construction for generating a sufficient but finite set of negative examples. For this reason, we consider language transduction tasks, where the inputs and outputs form two individual formal languages, and the goal is to learn a deterministic function that maps a word from one language to the other. Concretely, a deterministic finite transducer, defined as a 6-tuple $(Q, \Sigma_I, \Sigma_O, \delta, q_0, F)$, differs from a finite-state automaton in that its transition function $\delta : Q \times (\Sigma_I \cup \{\varnothing\}) \to (\Sigma_O \cup \{\varnothing\}) \times Q$ can output a symbol from the output alphabet $\Sigma_O$ (or the dummy token $\varnothing$) at every step. Thus, a transducer maps $w_I = a_1 a_2 \ldots a_n$ to $w_O = b_1 b_2 \ldots b_n$ if there exists a sequence of states $r_0, r_1, \ldots, r_n \in Q$ such that $r_0 = q_0$, $(b_{i+1}, r_{i+1}) = \delta(r_i, a_{i+1})$ for $i = 0, \ldots, n-1$, and $r_n \in F$. The transducer outputs $b_1 = \ldots = b_k = \varnothing$ while reading input word $w_I' = a_1 \ldots a_k \in \Sigma_I$ and outputs $w_O' = b_{k+1} \ldots b_n \in \Sigma_O$ while reading dummy tokens $a_{k+1} = \ldots = a_n = \varnothing$. Transducers therefore define a deterministic function $f : \Sigma_I^* \to \Sigma_O^*$, given by $f(w_I') = w_O'$, between two formal languages, which

is precisely what we want to learn. Thus, strictly speaking, we do not consider a hierarchy over formal languages but over language transduction tasks. Importantly, the machines required are identical to those in the Chomsky hierarchy, i.e., FSAs, PDAs, LBAs, and TMs (see Table 1), except that they can now also output values at each transition (i.e., the memory augmentations stay the same). We further discuss the transition from recognition to transduction, as well as prior approaches, in Appendix A.1.

## 4 METHODS

**Problem setup**  For each task we define input and output languages $L_I$ and $L_O$, with their corresponding alphabets $\Sigma_I$ and $\Sigma_O$. The goal of a task is to learn the deterministic function $f$ that maps words $\boldsymbol{x} \in L_I$ to words $\boldsymbol{y} \in L_O$, i.e., language transduction. To that end, we consider models $p_\theta(\cdot|\boldsymbol{x})$ as conditional distributions over next possible sequence continuations (with parameters $\theta$). We one-hot encode the tokens in both languages $L_I$ and $L_O$ as vectors of length $|\Sigma_I|$ and $|\Sigma_O|$, respectively, denoting the corresponding $j$th entries of the $i$th one-hot tokens $x_{ij}$ and $y_{ij}$. We use gradient-based training to minimize the average cross-entropy loss, where $C(\boldsymbol{x}, \boldsymbol{y}) := -\frac{1}{|\boldsymbol{y}|} \sum_i y_i^T \log p_\theta(\cdot|\boldsymbol{x})$ denotes the per-example loss (see Algorithm A.1). Note that we do not use auto-regressive models, but rather append $|\boldsymbol{y}|$ empty dummy tokens to the input. Thus, the model will know when the input sequence $\boldsymbol{x}$ has terminated by reading the first empty token. We compute the per-sequence accuracy as the percentage of correctly predicted tokens, i.e., $A(\boldsymbol{x}, \boldsymbol{y}) := \frac{1}{|\boldsymbol{y}|} \sum_i \mathbb{1} \left[ (\arg\max_j y_{ij}) = (\arg\max_j p_\theta(\cdot|\boldsymbol{x})_j) \right]$. Moreover, we compute the overall performance score as the per-sequence accuracy averaged over all sequences of unseen length. We run our experiments over 10 different network parameter initialization seeds and report the maximum score instead of the average, as this better indicates if an architecture is capable of learning the right algorithm *at all* (we provide the means and variances in Appendix B).

**Tasks**  For each level of the Chomsky hierarchy, we define various sequence prediction tasks ranging from modular arithmetic (R) to binary addition (CS). We place the most emphasis on CS tasks as they are currently out of reach for state-of-the-art models and thus provide an ideal benchmark to guide future research on architecture design. We list the tasks in Table 2 and provide a full formalization in Table A.1. Although seemingly simple, these tasks are well-known examples in the theory of computation and concisely demonstrate the key properties of the grammars in the Chomsky hierarchy, i.e., counting, repetition, long-distance dependencies, and hierarchy. Note that the levels of our tasks are independent of the levels of $L_I$ or $L_O$ but correspond to the automata that can solve the corresponding transduction tasks. For example, for the `Reverse String` task, $L_I = L_O = \{a, b\}^*$, which are both regular. However, learning the function $f$ that maps $w \in L_I$ to $reverse(w) \in L_O$ requires a deterministic push-down transducer, thus rendering the task deterministic context-free.

**Architectures**  We consider a wide range of neural network architectures, including both state-of-the-art and memory-augmented models (full details in Appendix A.2). Our goal is to have at least one model per level of the Chomsky hierarchy. To that end, we use a standard RNN as a controller and augment it with two kinds of differentiable memory structures: a deterministic stack and a bounded tape. The stack and tape have elements in $\mathbb{R}^d$ with $d = 8$, and we ensure that the stack and tape sizes are large enough to simulate infinite memory during training and testing. The Stack-RNN (Joulin & Mikolov, 2015) can perform any linear combination of PUSH, POP, and NO-OP operations on the stack. The Tape-RNN, which is inspired by the Baby-NTM (Suzgun et al., 2019b), can perform any linear combination of WRITE-LEFT, WRITE-RIGHT, WRITE-STAY, JUMP-RIGHT, JUMP-LEFT (details in Appendix A). In addition to these memory-augmented models, we also evaluate LSTMs (Hochreiter & Schmidhuber, 1997) and Transformers (Vaswani et al., 2017). We compute the Transformer output from the encoder only model that overlooks the whole input sequence. We consider five different positional encodings: none, classical sin/cos (Vaswani et al., 2017), the rotary positional encodings (RoPE) (Su et al., 2021), ALiBi (Press et al., 2021), and the relative positional encodings of Transformer-XL (Dai et al., 2019). The latter two have been shown to allow extrapolation to longer sequences on some language modeling tasks. All models receive the tokens $\boldsymbol{x}$ as an input and are expected to produce the output $\boldsymbol{y}$ (i.e., we do not feed the output back to the input unlike sequence-to-sequence models (Sutskever et al., 2014)). We study CNNs, Stack-LSTMs, NDStack-RNNs (DuSell & Chiang, 2020; 2022), and autoregressive versions of our models in Appendix B.

Table 2: Score (in percentage, see Section 4), i.e., accuracy averaged over all test lengths and maximized over 10 random seeds (and other hyperparameters, see Appendix A). We consider a model to generalize successfully (bold) if its score $\geq 90\%$. The random accuracy is $50\%$ (except for `Cycle Navigation` (R), `Bucket Sort` (CS), and the two modular arithmetic tasks where it is $20\%$). We observe that, in general, RNNs with more permissive memory structures can solve more challenging tasks. We denote permutation-invariant tasks with †, counting tasks with ⋆, tasks requiring a nondeterministic controller with ○, and tasks requiring superlinear running time with ×.

| Level | Task | RNN | Stack-RNN | Tape-RNN | Transformer | LSTM |
|---|---|---|---|---|---|---|
| R | Even Pairs | **100.0** | **100.0** | **100.0** | **96.4** | **100.0** |
| | Modular Arithmetic (Simple) | **100.0** | **100.0** | **100.0** | 24.2 | **100.0** |
| | Parity Check† | **100.0** | **100.0** | **100.0** | 52.0 | **100.0** |
| | Cycle Navigation† | **100.0** | **100.0** | **100.0** | 61.9 | **100.0** |
| DCF | Stack Manipulation | 56.0 | **100.0** | **100.0** | 57.5 | 59.1 |
| | Reverse String | 62.0 | **100.0** | **100.0** | 62.3 | 60.9 |
| | Modular Arithmetic | 41.3 | **96.1** | **95.4** | 32.5 | 59.2 |
| | Solve Equation○ | 51.0 | 56.2 | 64.4 | 25.7 | 67.8 |
| CS | Duplicate String | 50.3 | 52.8 | **100.0** | 52.8 | 57.6 |
| | Missing Duplicate | 52.3 | 55.2 | **100.0** | 56.4 | 54.3 |
| | Odds First | 51.0 | 51.9 | **100.0** | 52.8 | 55.6 |
| | Binary Addition | 50.3 | 52.7 | **100.0** | 54.3 | 55.5 |
| | Binary Multiplication× | 50.0 | 52.7 | 58.5 | 52.2 | 53.1 |
| | Compute Sqrt | 54.3 | 56.5 | 57.8 | 52.4 | 57.5 |
| | Bucket Sort†⋆ | 27.9 | 78.1 | 70.7 | **91.9** | **99.3** |

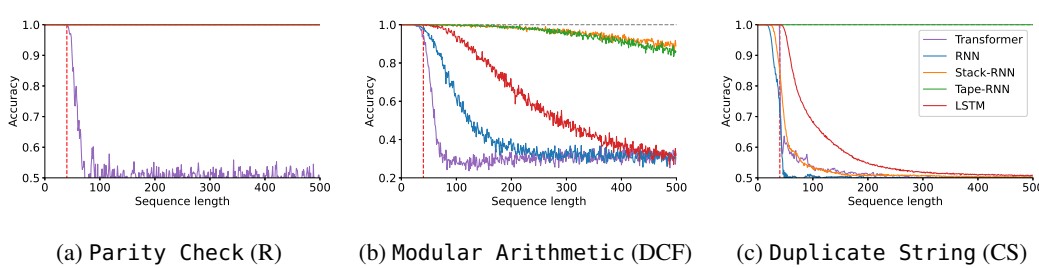

(a) `Parity Check` (R)     (b) `Modular Arithmetic` (DCF)     (c) `Duplicate String` (CS)

Figure 2: Performance curves on three tasks. The dashed vertical red line is the training range, meaning that sequences to the right have not been seen during training and thus measure generalization.

**Data generation** Instead of using fixed-size datasets, we define training and test distributions from which we continually sample sequences. We define the maximum training sequence length as the *training range* $N$, with $N = 40$. Every mini-batch is generated by first sampling a length $\ell$ from the uniform distribution $\mathcal{U}(1, N)$ and then sampling sequences of length $\ell$ from the task's generative grammar. For testing, we sample the sequence length $\ell$ from $\mathcal{U}(N + 1, M)$, with $M = 500$.

We provide an open-source implementation of our models, tasks, and training and evaluation suite at `https://github.com/deepmind/neural_networks_chomsky_hierarchy`.

## 5  RESULTS

In this section, we provide the results of our empirical evaluation. In Section 5.1 we relate the networks to the Chomsky hierarchy, in Section 5.2 we analyze the algorithms learned by the networks, and in Sections 5.3 and 5.4 we discuss the performance of LSTMs and Transformers in more detail.

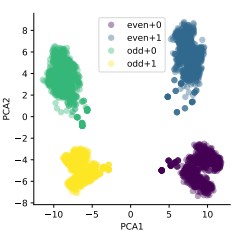

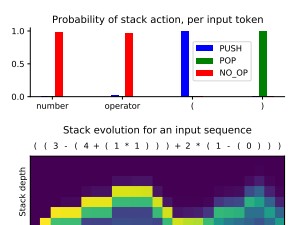

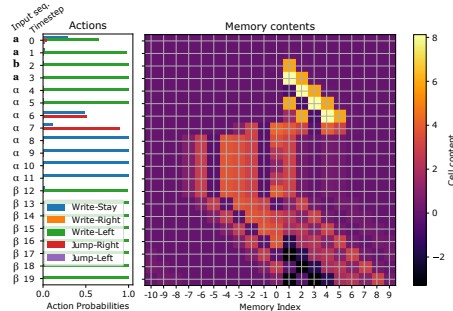

(a) Internal RNN states on `Parity Check` (R), colored by the previous state and the current input.

(b) Stack-RNN action probabilities (*above*) and stack values (*below*) on the `Modular Arithmetic` (DCF) task.

(c) Tape-RNN action probabilities and memory values on `Duplicate String` (CS).

Figure 3: Analysis of the internal representations and memory dynamics of an RNN on a regular task, a Stack-RNN on a DCF task, and a Tape-RNN on a CS task. The reverse-engineered network mechanisms are indicative of the correct algorithms that solve the corresponding tasks.

## 5.1 MAIN RESULT

We evaluate all architectures (defined in Section 4) on all tasks (defined in Table A.1) and show the generalization accuracy averaged over the segment $\mathcal{U}(N+1, M)$ (as described in Section 4) in Table 2. Furthermore, Fig. 2 displays the accuracy per sequence length for the best-performing model per architecture on three sample tasks (see Appendix B for the remaining tasks). As the test sequences have not been seen during training, we can gauge whether the architectures can learn the "correct" algorithm. We observe that the networks generally match the computational models associated with the Chomsky hierarchy: RNNs can solve tasks up to the regular level, Stack-RNNs up to the DCF level, and Tape-RNNs up to the CS level. However, the match is not perfect, as some architectures cannot solve tasks at their supposed level, e.g., Stack-RNN cannot solve `Solve Equation` (DCF) and the Tape-RNN cannot solve the (challenging) `Binary Multiplication` (CS) and `Compute Sqrt` (CS) tasks. This may be due to shortcomings in the architecture, the training protocol, or the particular difficulty of the tasks. For example, `Solve Equation` (DCF) is context-free but requires a nondeterministic FSA controller because it has to compute multiple modular arithmetic expressions for the different values of the unknown variable $x$ and compare their results to the actual solution of the equation, which is only provided at the end of the input sequence. Similarly, `Binary Multiplication` (CS) is a context-sensitive task but has a quadratic complexity, which is infeasible for the architectures we investigate. Conversely, some architectures show non-trivial performance on tasks beyond their supposed level, such as the Stack-RNN on `Bucket Sort` (CS) (more details in Appendix B). Finally, we observe that Transformers and LSTMs are not well-aligned with the Chomsky hierarchy: Transformers fail on regular tasks, while LSTMs can solve (counting) tasks more difficult than regular. We discuss both cases in detail below (Sections 5.3 and 5.4).

## 5.2 ANALYSIS OF LEARNED PROGRAM STRUCTURE

To provide further evidence supporting the claim that some of the networks are able to learn the right algorithm, we now provide an analysis of solutions learned by some of our networks on our tasks, by investigating internal state representations and memory-update dynamics.

**Regular tasks** According to the Chomsky hierarchy, we expect networks that solve regular tasks to simulate a finite-state automaton. We investigate this hypothesis by analyzing the internal states of the RNN controller and checking whether they are finite (or form a finite number of clusters). Figure 3a shows a scatter plot of a PCA projection of the internal RNN states when trained on `Parity Check` (R). We observe that the states form four clusters, each representing one of the four values of the tuple (subsequence result, last token seen). In principle, one can solve this task with only two states, corresponding to the subsequence result: If the state is 1 and the input token is 1, move to state 0, otherwise stay in state 1, and finally output the state itself. However, due to the problem symmetry, the network also learns to compute the parity of the number of zeros in the string, therefore adding another two states to the computation. We also investigate whether and how Stack-RNNs use their

stack for this task, which is unnecessary, in principle. Indeed, the Stack-RNNs only use NO-OP or POP (which is equivalent to NO-OP in the absence of PUSH) actions, indicating that the RNN controller uses only a finite-state automaton and does not rely on the stack (see Appendix B.3 for more details).

**Deterministic context-free tasks**   On DCF tasks, we anticipate Stack-RNNs to use their stack. Indeed, on Reverse String (DCF), the networks learn the expected algorithm: PUSH until an empty token appears, and then POP until the stack is empty. Moreover, the internal state of the controller seems to be largely guided by the last input token and not the last state (details in Appendix B.3). On Modular Arithmetic (DCF), which requires a stack to remember past results when entering a new bracket, the networks use the stack exactly as expected. Figure 3b shows that the Stack-RNN learns to solve this task by: (i) pushing the last result when a bracket is opened, (ii) not operating on the stack and only using its internal controller when a number or an operator is seen, and (iii) popping the last result from the stack when a bracket is closed. The top plot reports the probability of each action given an input, and the bottom plot shows the stack evolution along the sequence. On the bottom plot, the x-axis is the timestep and the y-axis is the cell position in the stack (top of the stack on the bottom). Each cell is colored according to the first principal component of its actual content.

**Context-sensitive tasks**   In contrast to the RNN and Stack-RNN, the reverse-engineered network mechanisms for the Tape-RNN are less interpretable. Nevertheless, Fig. 3c shows that the Tape-RNN has learned an algorithm that is indicative of the data-generating grammar of the Duplicate String (CS) task. The input to the network consists of the sequence $a, a, b, a$ of length $\ell = 4$, followed by $2\ell$ $\alpha$-symbols that the RNN can use to compute and manipulate the tape, followed by $2\ell$ $\beta$-symbols used to output the result (i.e., the sequence $a, a, b, a, a, a, b, a$ of length $2\ell$). The panels show the action probabilities and memory contents over time. We observe that up to time $t = 5$ the RNN writes on the memory with WRITE-LEFT. Then, on time-step 6 the RNN mixes JUMP-RIGHT and WRITE-STAY to duplicate the input string in the memory, before moving back to the initial position with JUMP-RIGHT on time-step 7. After this, the RNN idles using WRITE-STAY until it reaches the first $\beta$-symbol, whereupon it starts outputting the result using WRITE-LEFT actions.

### 5.3   LSTMs

LSTMs are similar to RNNs as they also carry a hidden state and are unrolled over a sequence of inputs. However, prior work showed that LSTMs are strictly more powerful than RNNs because they can also solve counting tasks (Bodén & Wiles, 2000; 2002; Gers & Schmidhuber, 2001; Merrill, 2019; Weiss et al., 2018). Indeed, Table 2 provides further evidence for this claim since LSTMs are capable of solving Bucket Sort (CS) almost perfectly, even though the task is context-sensitive. That is, counting tasks can reside on levels higher than regular in the Chomsky hierarchy, since every Turing machine can, in theory, be simulated by a two-counter machine (Minsky, 1967). However, finding an architecture that solves non-regular tasks with counters via gradient descent is more difficult than when the controller has access to a more permissive memory structure, such as a stack or a tape.

### 5.4   TRANSFORMERS

Unlike the other architectures, Transformers are *permutation invariant* w.r.t. the position of a token in a sequence because they rely on a global attention mechanism. As a result, Transformers are more scalable since all tokens can be processed in parallel, however, at the cost of reduced expressivity (i.e., permutation invariance). Similar to real-world problems, most of our tasks are not permutation invariant, e.g., reversing a string depends on the tokens' position. To overcome this problem, Transformers are generally augmented with positional encodings, which are added to the tokens to simulate some notion of position. Consequently, the augmented tokens intrinsically contain position information, i.e., the token $x$ at position 0 will no longer be considered the same as the same token $x$ at position 10. As described in Section 4, we evaluate five encodings (none, sin/cos, RoPE, ALiBi, and the relative positional encoding from Transformer-XL) and report the best-performing variant.

Indeed, Table 2 shows that Transformers are most successful on permutation-invariant tasks: They solve Bucket Sort (CS) and demonstrate non-trivial generalization on Cycle Navigation (R). Moreover, they solve Even Pairs (R), which is not permutation-invariant but only requires checking whether the first and last tokens are equal (i.e., all other positions are irrelevant). However, for all other tasks, Transformers fail to generalize, regardless of their positional encodings (see Fig. 4a for the

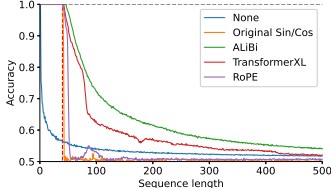 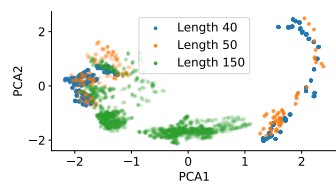

(a) Accuracy per length by positional encoding.  (b) First layer activations (PCA) by sequence length.

Figure 4: Transformers on `Reverse String` (DCF). Without positional encodings, Transformers are permutation invariant, and with encodings, positions are out-of-distribution for longer sequences.

different positional encodings on `Reverse String` (DCF) and Fig. B.7 for the remaining tasks). In particular, Transformers are unable to solve all permutation-invariant tasks (i.e., `Parity Check` (R), which is a well-known failure mode (Chiang & Cholak, 2022)). We hypothesize that this poor performance is due to the positional encodings, which take on new values when the sequences grow longer (even for relative encodings), meaning that the network activations will be out-of-distribution. Indeed, Fig. 4b shows that the 2D PCA of the first layer activations is similar for sequences of length 40 (i.e., the training range, in blue) and 50 (in orange), but completely different for length 150 (in green). All our other models, which do not rely on global self-attention, do not suffer from this effect, as they only have access to a local view of the input token (and, potentially, a local memory reading).

# 6 DISCUSSION

In this paper, we considered generalization from a sequence prediction viewpoint, using formal language theory to establish the computational complexity of each task. Naturally, we had to choose a maximum test sequence length for our evaluations, which theoretically renders all of our input and output languages finite. However, our sequences share the structure of an infinite language, allowing us to measure generalization by testing on input sequences that are significantly longer than those seen during training (and are thus out-of-distribution). Therefore, observing successful generalization in our setting is strong evidence that the network has learned the "correct" algorithm. However, for some tasks where we consider generalization to be successful we still see a slight degradation of the accuracy as the test length increases. This is a result of implementing finite state machines and memory updates with a neural network trained via SGD: Even slight numerical inaccuracies in state-transitions or memory dynamics might accumulate when increasing the test sequence length.

Our experiments indicate that RNNs, LSTMs, and Transformers, admittedly powerful architectures, are fairly limited in terms of their ability to generalize to longer inputs. However, this result must be contrasted with their ability to generalize and detect/extrapolate patterns for fixed size inputs or fixed size context windows. Transformers, for instance, are capable of learning complex and highly structured generalization patterns, but they cannot overcome the limitation of not having an extendable memory (which only becomes apparent when probed as in our experiments). This might imply hard limits for scaling laws (Kaplan et al., 2020): Even significantly increasing the amount of training data and the size of a Transformer are insufficient for it to "climb the Chomsky hiearchy".

# 7 CONCLUSION

We leveraged the theory of computation to better understand how and why neural networks generalize on algorithmic sequence prediction tasks. Our extensive empirical evaluation demonstrates that there is a model hierarchy on the tasks we investigated, which are representative of the different levels of the Chomsky hierarchy. In particular, we showed that state-of-the-art architectures, such as LSTMs and Transformers, cannot solve seemingly simple tasks, such as duplicating a string, when evaluated on sequences that are significantly longer than those seen during training. Moreover, we showed that models interacting with an external memory structure, such as a stack or a finite tape, can climb the Chomsky hierarchy, indicating a promising direction for improvements in architecture design.

## 8   ETHICS STATEMENT & LIMITATIONS

Although our results are consistent with prior theoretical analyses (Ackerman & Cybenko, 2020; Merrill, 2019; Weiss et al., 2018), and suggest that we can group neural architectures according to the Chomsky hierarchy, our claims are limited to our empirical study. In particular, we cannot guarantee that no tasks exist higher in the hierarchy (Fig. 1) that an architecture could solve. Similarly, we cannot guarantee that no tasks exist lower or on the same level of the hierarchy that an architecture cannot solve. Moreover, without extracting and analyzing the automata implemented by the networks, we cannot strictly claim that our architectures generalize to arbitrary-length inputs, as we only test up to a maximum length. Finally, our results are w.r.t. our precise experimental setting, i.e., if an architecture fails to generalize we cannot guarantee that no weight configuration would solve the task; we were simply unable to find such a configuration with our training protocol.

### ACKNOWLEDGMENTS

We thank Ann He, Chris Dyer, Markus Kunesch, Róbert Csordás, Tom McGrath, and Zhengdong Wang for their helpful feedback and insightful conversations.

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

## A  EXPERIMENTAL DETAILS

### A.1  LANGUAGE RECOGNITION VS. TRANSDUCTION

In practice, learning to recognize a formal language by training a classifier on words and non-words of a language is hard because there is, in general, no formal construction for generating a sufficient but finite set of negative examples. Thus, prior work has relied on other approaches to evaluate language recognition capabilities. These approaches are variants of language generation, which are as hard as their recognition counterpart (for a given language) and thus are good proxy tasks.

The first approach is autoregressive token prediction (standard in the machine learning community (Bahdanau et al., 2015; Vaswani et al., 2017)): Given a prefix $p$ of a word $w \in L$, predict the probability of its next token $t \in \Sigma$. The main problem with this approach is that the probabilities depend on the length distribution of the training strings. For instance, for the language of palindromes of length 2, with alphabet $\Sigma = \{a, b\}$, the output probabilities are $P(a|\text{prefix} = a) = 1$ and $P(b|\text{prefix} = b) = 1$. However, for palindromes of length 3, the probabilities are very different, i.e., $P(a|\text{prefix} = a) = 0.5$ and $P(b|\text{prefix} = b) = 0.5$. Therefore, there is a conflict between the different distributions, and no model can solve the task without knowing the length beforehand. Accordingly, increasing the length at test time without informing the model makes it impossible to find the correct new probabilities and, therefore, to generalize to inputs of unseen lengths.

The second approach is to predict the set of possible next tokens and not their probabilities (Bhattamishra et al., 2020; Suzgun et al., 2019a). In this case, the distribution of possible next tokens does not change with the length, and increasing the length at test time is possible. However, for each prefix, the target will consists of a set of tokens, which is incompatible with the (standard sequence) prediction setting and thus of lesser interest to the wider machine learning community.

Our approach is similar to the second one, but we only consider prefixes for which there is only one possible (deterministic) continuation. Thus, instead of language generation, we consider language transduction: The inputs and outputs form two individual languages, and the goal is to learn the deterministic function that maps a word from one to the other. The automata solving these tasks are the classical automata from Fig. 1 but with outputs attached to each transition and are denoted transducers. Therefore, the hierarchy that we consider is a hierarchy over transducers and not over acceptors as detailed in Section 3. However, the two types of machines (acceptor automata and transducer automata) are interchangeable for the purposes of our investigation as they are both memory-augmented finite-state automata that reside on different levels of the Chomsky hierarchy depending on their type of external memory access (none, a stack, or a tape).

### A.2  MODELS

Here we describe our neural architectures in more detail. All models have a final linear readout layer to produce the output logits. We do not use any regularization, e.g., weight decay, dropout (except for the Transformer models), etc. For training of the memory-augmented models, we set the tape size to 256 and the stack size to 128 and manually increase their sizes when testing on longer sequences. Thus, as these memory sizes are significantly larger than the training range $N$ (see Section 4), the finite memory models appear infinite for our neural architectures in the context of our experimental evaluation. Moreover, increasing the memory size does not affect how the RNN controller takes actions, since it only has access to a local view of the external memory (e.g., the top of the stack or the current memory cell).

**RNN**  A vanilla single-layer RNN (Elman, 1990) with hidden size of 256.

**Stack-RNN**  A single-layer RNN controller of hidden size 256 with access to a differentiable stack (Joulin & Mikolov, 2015). The controller can perform any linear combination of PUSH, POP, and NO-OP on the stack, with action weights given by a softmax over a linear readout of the RNN output. Each cell of the stack contains a real vector of dimension 8.

**NDStack-RNN**  A single-layer RNN controller of hidden size 256 with access to a differentiable nondeterministic stack. We consider the NDStack-RNN proposed by (DuSell & Chiang, 2020; 2022), which simulates a nondeterministic stack via dynamic programming. Concretely, we use normalized

actions and nondeterministic state reading with 2 states and sweep over $2, 4$ symbols. We did not use the unnormalized actions as suggested in (DuSell & Chiang, 2022).

**Tape-RNN**   A single-layer RNN controller of hidden size 256 with access to a differentiable tape, inspired by the Baby-NTM architecture (Suzgun et al., 2019b). The controller can perform any linear combination of WRITE-RIGHT, WRITE-LEFT, WRITE-STAY, JUMP-LEFT, and JUMP-RIGHT on the tape, with action weights given by a softmax. The actions correspond to: writing at the current position and moving to the right (WRITE-RIGHT), writing at the current position and moving to the left (WRITE-LEFT), writing at the current position (WRITE-STAY), jumping $\ell$ steps to the right without writing (JUMP-RIGHT), where $\ell$ is the length of the input, and jumping $\ell$ steps to the left without writing (JUMP-LEFT). Finally, to allow the Tape-RNN to perform memory operations without having to produce the output, we append "computation" tokens (different from the empty tokens $\varnothing$ used to produce the output, see Appendix A.4 below) to the input sequence. As in the Stack-RNN, each cell of the tape contains a real vector of dimension 8.

**LSTM**   A vanilla single-layer LSTM Hochreiter & Schmidhuber (1997) of hidden size 256.

**Stack-LSTM**   A single-layer LSTM controller of hidden size 256 with access to a differentiable stack Joulin & Mikolov (2015). The only difference to the Stack-RNN is that this model uses an LSTM as the controller.

**Transformer encoder**   A vanilla Transformer encoder (Vaswani et al., 2017). We do not use the autoregressive sequence-to-sequence model to produce the output but attend the whole input (including the empty tokens $\varnothing$). We use five blocks with $d_{model} = 64$, where each block is composed of an attention layer, two dense layers, and a layer normalization. We add a residual connections as in the original architecture (Vaswani et al., 2017). We consider five different positional encodings: none, classical sin/cos (Vaswani et al., 2017), RoPE (Su et al., 2021), ALiBi (Press et al., 2021), and the relative positional encoding from Transformer-XL (Dai et al., 2019).

**Transformer (autoregressive)**   A vanilla Transformer (Vaswani et al., 2017). We do not use empty tokens $\varnothing$ anymore, and the output is produced via the decoder (using causal masking). We use the same hyperparameters and positional encodings as for the Transformer encoder above.

**Convolutional Neural Network**   A vanilla CNN, consisting of 5 convolutional layers with 32 channels each and a final MLP that produces the task output(s). We compute 2D convolutions over the time and (one-hot encoding) channel dimension. That is, the first kernel will consider the entire one-hot encoding at once for every time step. For all other kernel dimensions (time and filters), we sweep over the values $\{3, 5, 7, 11\}$.

## A.3   TASKS

Here, we describe our tasks (listed in Table A.1) in more detail. As mentioned in Section 4, all tasks consist of an input sequence $\boldsymbol{x} \in L_I$ from which the models have to predict the target sequence $\boldsymbol{y} \in L_O$. We append $|\boldsymbol{y}|$ empty tokens $\varnothing$ to the input sequence, such that the models can process the entire input sequence before having to produce the output (i.e., they know the input length before having to produce the output). Note that half of our tasks have an output length of 1 and are thus just standard classification tasks with the set of classes being the alphabet of the output language. The networks still see one empty token $\varnothing$ at the end of the input string to signify that the input phase has finished and that the output must be computed. We do not consider NDCF tasks since the distinction between DCF and NDCF is fairly technical and does not fundamentally change the type of memory structure required (i.e., a stack).

**Even Pairs (R)**   Given a binary sequence, e.g., $\boldsymbol{x} = aabba$, compute if the number of $ab$s and $ba$s is even. For $aabba$, we have one $ab$ and $ba$, meaning that the total number is 2 and thus even, i.e., $\boldsymbol{y} = b$ (where we arbitrarily equate odd with $a$ and even with $b$). This task is equivalent to computing whether the first and last character of the string are equal. Therefore, the task is regular since we can solve it with a 2-state finite-state machine that looks at the first bit of the sequence and compares it with the last one.

**Modular Arithmetic (Simple) (R)**  Given a sequence of numbers in $\{0, 1, 2, 3, 4\}$ and operations in $\{+, -, \cdot\}$, compute the result modulo 5. For example, $\boldsymbol{x} = 1 + 2 - 4$ evaluates to $\boldsymbol{y} = 4$. Note that the length of the input sequences must be of odd. Therefore, if we sample an even length during the training/evaluation process, we return an input sequence of this length plus 1. This task is regular since we can solve it with a 5-state finite-state machine, which transitions for each new operation with the state being equal to the current result.

**Parity Check (R)**  Given a binary string, e.g., $aaabba$, compute if the number of $b$s is even. The sequence $\boldsymbol{x} = aaabba$ contains 2 $b$s, which is even, i.e., $\boldsymbol{y} = b$ (where we arbitrarily equate odd with $a$ and even with $b$). This task is regular since we can solve it with a 2-state finite-state machine, which transitions every time it sees a token different from its current state.

**Cycle Navigation (R)**  Given a sequence of movements on a cycle of length 5, compute the end position. The movements are STAY, INCREASE, DECREASE and are represented as $\{0, 1, 2\}$. The agent always starts at position 0. For example, $010211$ means the agent stops at position $0 + 0 + 1 + 0 - 1 + 1 + 1 = 2$, and the output class is 2. This task is regular since we can solve it with a 5-state finite-state machine, similar to `Modular Arithmetic (Simple)` (R), as the task can be converted into a sum modulo 5.

**Modular Arithmetic (DCF)**  Given a sequence of numbers in $\{0, 1, 2, 3, 4\}$, brackets, and operations in $\{+, -, \cdot\}$, compute the result modulo 5. For example, the sequence $\boldsymbol{x} = -(1-2)\cdot(4-3\cdot(-2))$ evaluates to $\boldsymbol{y} = 0$. Note that we do not have the same problem of odd lengths as in `Modular Arithmetic (Simple)` (R), since we can use substrings of the form $(-z)$ with $z \in \{0, 1, 2, 3, 4\}$, which have length 4 and thus can be used to produce input strings of even length. The task is DCF since we can solve it with a 5-state finite-state machine and an external stack. That is, similar to the `Modular Arithmetic (Simple)` (R) task, the finite-state machine transitions for each new operation, with the state being equal to the current result. Moreover, the stack is used to handle brackets: The machine pushes if the current input token is an opening bracket, pops if it is a closing bracket, and uses no-ops to let the finite-state machine work otherwise.

**Reverse String (DCF)**  Given a binary string, e.g., $\boldsymbol{x} = aabba$, compute its reverse, i.e., $\boldsymbol{y} = abbaa$. The task is DCF since we can solve it by pushing the tokens of the input sequence on a stack until the first empty token $\varnothing$ appears and then popping the tokens from the stack, outputting the values one by one. This task cannot be solved with a finite-state machine since the number of possible outputs is infinite.

**Solve Equation (DCF)**  Given an equation consisting of numbers in $\{0, 1, 2, 3, 4\}$, brackets, operations in $\{+, -, \cdot\}$, and an unknown variable $z$, compute the value of $z$ such that the equation holds modulo 5. For example, $\boldsymbol{x} = -(z - 2) \cdot (4 - 3 \cdot (-2)) = 0$ modulo 5 holds for $z = 1$, i.e., $\boldsymbol{y} = 1$. This task is DCF since we can solve it by simulating all possible values of $x$ in a nondeterministic finite-state machine and producing each result using the same algorithm as in `Modular Arithmetic` (DCF). Note that a nondeterministic finite-state machine can be simulated with a deterministic one, implying that this task is DCF and not NDCF.

**Stack Manipulation (DCF)**  Given a binary string representing a stack's content (given bottom-to-top) and a sequence of PUSH $a$, PUSH $b$, or POP actions, execute the actions on the stack and return the final stack content (in a top-to-bottom manner, i.e., as if popping the resulting stack). For example, executing the sequence POP PUSH $a$ POP on the stack $abbaa$, i.e., $\boldsymbol{x} = abbaa$ POP PUSH $a$ POP, results in $\boldsymbol{y} = abba$. If a POP action is called on an empty stack, the action is ignored. This task requires stack manipulation by design and is therefore DCF.

**Binary Addition (CS)**  Given two binary numbers, e.g., 10010 and 101, compute their sum in base 2, i.e., $\boldsymbol{y} = 10111$ for $\boldsymbol{x} = 10010 + 101$. The inputs and outputs are given in little-endian representation. The task is CS since it exhibits cross-serial dependencies because the two numbers are provided one after the other (and not bit-by-bit at the same time).

**Binary Multiplication (CS)**  Given two binary numbers, e.g., 100 and 10110, compute their product in base 2, i.e., $\boldsymbol{y} = 1011000$ for $\boldsymbol{x} = 100 * 10110$. The inputs and outputs are given in

Table A.1: Our tasks with their level in the Chomsky hierarchy and example input/output pairs. The †
denotes permutation-invariant tasks; the ⋆ denotes counting tasks; the ∘ denotes tasks that require a
nondeterministic controller; and the × denotes tasks that require superlinear running time in terms of
the input length.

| Level | Name | Example Input | Example Output |
|-------|------|---------------|----------------|
| R | Even Pairs | $aabba$ | True |
| | Modular Arithmetic (Simple) | $1 + 2 - 4$ | 4 |
| | Parity Check† | $aaabba$ | True |
| | Cycle Navigation† | 011210 | 2 |
| DCF | Stack Manipulation | $abbaa$ POP PUSH $a$ POP | $abba$ |
| | Reverse String | $aabba$ | $abbaa$ |
| | Modular Arithmetic | $-(1-2) \cdot (4 - 3 \cdot (-2))$ | 0 |
| | Solve Equation∘ | $-(x-2) \cdot (4 - 3 \cdot (-2))$ | 1 |
| CS | Duplicate String | $abaab$ | $abaababaab$ |
| | Missing Duplicate | 10011021 | 0 |
| | Odds First | $aaabaa$ | $aaaaba$ |
| | Binary Addition | $10010 + 101$ | 10111 |
| | Binary Multiplication× | $10010 * 101$ | 1001000 |
| | Compute Sqrt | 100010 | 110 |
| | Bucket Sort†⋆ | 421302214 | 011222344 |

little-endian representation. The task is CS since it exhibits cross-serial dependencies because the
two numbers are provided one after the other (and not bit-by-bit at the same time).

**Compute Sqrt (CS)**   Given a binary number, e.g., 101001, compute the floor of its square root, i.e.,
$y = \lfloor \sqrt{101001} \rfloor = 101$. The inputs and outputs are given in little-endian representation. This task
is CS since the digit-by-digit algorithms to compute the integer square root require many sequence
manipulations, including binary multiplications, which is a CS task.

**Duplicate String (CS)**   Given a binary string, e.g., $\boldsymbol{x} = abaab$, output the string twice, i.e., $\boldsymbol{y} = abaababaab$. This task is CS since it corresponds to the well-known language $\{ww \mid w \text{ is a word}\}$
(cf. Section 3), which is CS.

**Missing Duplicate (CS)**   The input is a binary string of the form $ww$ where $w$ is itself a binary
string. One token in this string has been hidden, and the network must find out which one it is,
by looking at the value at the same position but on the other side of the string. For instance, if
$\boldsymbol{x} = ab\_aba$ (i.e., $w = aba$), then $\boldsymbol{y} = a$. This task is CS since it requires on the recognition of the
well-known language $\{ww \mid w \text{ is a word}\}$ (cf. Section 3), which is CS.

**Odds First (CS)**   Given a binary string $t_1...t_n$, output $t_1 t_3 t_5 ... t_2 t_4 t_6 ....$ For example, the output
corresponding to $\boldsymbol{x} = aaabaa$ is $\boldsymbol{y} = aaaaba$. The task is CS because it exhibits cross-serial
dependencies (the output is the interleaved input).

**Bucket Sort (CS)**   Given a string over an alphabet of fixed size (5 in our case), return the sorted
string. Since the alphabet has a fixed size, the task can be solved via bucket sort, which only requires
a finite amount of counters (i.e., 5 counters in our case). For example, the output corresponding to
$\boldsymbol{x} = 421302214$ is $\boldsymbol{y} = 011222344$. The task is CS because it requires multiple counters that keep
track of the number of occurrences for every token.

## A.4 PROBLEM SETUP

We now provide the full formalization of our problem setup (described in Section 4). Algorithm A.1 illustrates our training procedure. We train on sequences of length $\ell \sim \mathcal{U}(1, N)$, with $N = 40$ (using batches of size 128). For testing, we consider sequences of length sampled from $\mathcal{U}(N + 1, M)$, with $M = 500$. In general, we report the score, which we define as the per-sequence accuracy $A$ (see Section 4) averaged over all sequences of unseen length, i.e., score $:= \frac{1}{M-N} \sum_{\ell=N+1}^{M} A(\boldsymbol{x}, \boldsymbol{y})$. We use the Adam optimizer (Kingma & Ba, 2015) with default hyperparameters for $1\,000\,000$ steps, which we have found to be sufficient to achieve a near-optimal training performance for all tasks and architectures. As described in Algorithm A.1, we augment our input sequences $\boldsymbol{x}$ with $m$ empty tokens $\varnothing$, where $m$ is the length of the corresponding target sequence $\boldsymbol{y}$, so that the networks only need to start outputting after they have consumed the entire input sequence. Moreover, we add computational tokens (which are different from the empty tokens $\varnothing$) to the input sequence $\boldsymbol{x}$ for the Tape-RNN to enable it to perform memory operations without having to output the result. In general, we run all experiments with 10 different random seeds (used for network parameter initialization) and three learning rates ($1 \times 10^{-4}$, $3 \times 10^{-4}$ and $5 \times 10^{-4}$), and we report the result obtained by the hyperparameters with the maximum score (means and standard deviations in Appendix B). However, for Transformers, NDStack-RNNs, and Tape-RNNs we sweep over additional hyperparameters: For Transformers we consider five different positional encodings, for the NDStack-RNN we optionally read from internal states of the nondeterministic stack (one of the improvements proposed by DuSell & Chiang (2022)) and sweep over 2 and 4 symbols, and for the Tape-RNN we consider various numbers of computation tokens ($0$, $\ell$, or $2\ell$), and various numbers of tapes (1 and 4).

---

**Algorithm A.1:** Training pipeline for our sequence prediction tasks. The comments (in blue) show an example output for the `Reverse String` (DCF) task.

**Input:** model $p_\theta(\cdot|\boldsymbol{x})$ with parameters $\theta$, learning rate $\alpha$, number of training steps $S$

1   Initialize parameters $\theta$
2   **for** $i \leftarrow 1$ **to** $S$ **do**
3      Sample length $\ell$ from $\mathcal{U}(1, 40)$             /* $\ell = 3$ */
4      Sample sequence $\boldsymbol{x}$ of length $\ell$ from the task's input grammar     /* $\boldsymbol{x} = 011$ for $\ell = 3$ */
5      Compute the corresponding output sequence $\boldsymbol{y}$ of length $m$     /* $\boldsymbol{y} = 110$ with $m = 3$ */
6      Pad $\boldsymbol{x}$ with $m$ empty tokens $\varnothing$ such that the model only needs to start outputting *after* having consumed the entire input sequence, i.e., $\boldsymbol{x} \leftarrow \boldsymbol{x} \cdot \varnothing \ldots \varnothing$     /* $\boldsymbol{x} = 011\varnothing\varnothing\varnothing$ */
7      Set the output to empty sequence of length $m$, i.e., $\boldsymbol{o} \leftarrow \varnothing \ldots \varnothing$
8      **for** $t \leftarrow 1$ **to** $m$ **do**
9          Compute the output probability distribution $o_t \leftarrow p_\theta(\cdot|\boldsymbol{x}_{1:\ell+t})$     /* $o_t \in \mathbb{R}^2$ */
10      **end**
11      Compute the cross-entropy loss (averaged over output tokens) $\mathcal{C} \leftarrow -\frac{1}{m} \sum_{t=1}^{m} y_t^\top \log o_t$
12      Update the parameters with gradient descent $\theta \leftarrow \theta - \alpha \nabla \mathcal{C}$
13      Compute the per-sequence accuracy $A \leftarrow \frac{1}{m} \sum_{t=1}^{m} \mathbb{1}\left[\arg\max_j y_{tj} = \arg\max_j o_{tj}\right]$
14   **end**

---

## A.5 COMPUTATIONAL RESOURCES

We implemented our evaluation suite in JAX (Bradbury et al., 2018) using the DeepMind JAX ecosystem (Babuschkin et al., 2020; Hessel et al., 2020; Hennigan et al., 2020). We make all of our code publicly available at `https://github.com/deepmind/neural_networks_chomsky_hierarchy`. We ran each task-architecture-hyperparameter triplet on a single TPU on our internal cluster. For RNNs, Stack-RNNs, LSTMs and Stack-LSTMs, the hyperparameters we have used are learning rates and seeds, which gives 15 (tasks) $\cdot$ 4 (models) $\cdot$ 3 (learning rates) $\cdot$ 10 (seeds) $= 1800$ TPU-units. For Tape-RNNs, the hyperparameters we have used are the learning rate, the number of computation steps tokens, the number of tapes, and the random seeds, which yields 15 (tasks) $\cdot$ 3 (learning rates) $\cdot$ 3 (computation steps) $\cdot$ 2 (tapes) $\cdot$ 10 (seeds) $= 2700$ TPU-units. For Transformers (encoder and autoregressive), the hyperparameters we have used are the learning rate, the positional encodings, and the random seeds, which gives 15 (tasks) $\cdot$ 2 (models) $\cdot$ 5 (positional encodings) $\cdot$ 3 (learning rates) $\cdot$ 10 (seeds) $= 4500$ TPU-units. For the NDStack-

Table A.2: Mean and standard deviation of the running times (in hours) for all the architectures and tasks. The † denotes permutation-invariant tasks; the ⋆ denotes counting tasks; the ○ denotes tasks that require a nondeterministic controller; and the × denotes tasks that require superlinear running time in terms of the input length.

| Level | Task | RNN | Stack-RNN | NDStack-RNN | Tape-RNN | Transformer | LSTM | Stack-LSTM | Autoreg. Transformer |
|---|---|---|---|---|---|---|---|---|---|
| R | Even Pairs | $0.37 \pm 0.01$ | $0.56 \pm 0.01$ | $8.50 \pm 0.16$ | $3.31 \pm 2.41$ | $1.19 \pm 0.73$ | $0.75 \pm 0.03$ | $2.97 \pm 0.01$ | $1.61 \pm 0.27$ |
| | Modular Arithmetic (Simple) | $0.57 \pm 0.04$ | $0.92 \pm 0.06$ | $8.34 \pm 0.19$ | $3.44 \pm 2.48$ | $1.31 \pm 0.55$ | $0.62 \pm 0.00$ | $3.05 \pm 0.01$ | $1.51 \pm 0.24$ |
| | Parity Check† | $0.59 \pm 0.02$ | $0.78 \pm 0.05$ | $8.54 \pm 0.12$ | $3.37 \pm 2.32$ | $1.23 \pm 0.58$ | $0.68 \pm 0.06$ | $2.96 \pm 0.01$ | $1.61 \pm 0.27$ |
| | Cycle Navigation† | $0.60 \pm 0.03$ | $0.57 \pm 0.07$ | $8.42 \pm 0.23$ | $3.26 \pm 2.25$ | $1.23 \pm 0.79$ | $0.76 \pm 0.03$ | $2.97 \pm 0.01$ | $1.66 \pm 0.45$ |
| DCF | Stack Manipulation | $4.07 \pm 0.12$ | $4.69 \pm 0.85$ | $17.80 \pm 1.43$ | $8.27 \pm 2.93$ | $5.06 \pm 0.65$ | $4.27 \pm 0.10$ | $8.89 \pm 0.04$ | $7.25 \pm 2.24$ |
| | Reverse String | $0.50 \pm 0.02$ | $0.88 \pm 0.11$ | $16.66 \pm 0.99$ | $4.58 \pm 2.88$ | $1.62 \pm 0.78$ | $0.70 \pm 0.03$ | $5.48 \pm 0.01$ | $4.32 \pm 2.03$ |
| | Modular Arithmetic | $2.70 \pm 0.08$ | $3.11 \pm 0.95$ | $9.04 \pm 0.78$ | $5.44 \pm 2.16$ | $3.59 \pm 0.65$ | $3.23 \pm 0.72$ | $5.21 \pm 0.04$ | $3.99 \pm 0.47$ |
| | Solve Equation○ | $2.82 \pm 0.06$ | $3.24 \pm 0.79$ | $9.16 \pm 0.64$ | $5.69 \pm 2.30$ | $3.59 \pm 0.49$ | $2.96 \pm 0.10$ | $5.25 \pm 0.05$ | $3.91 \pm 0.28$ |
| CS | Duplicate String | $0.57 \pm 0.05$ | $1.33 \pm 0.13$ | $25.25 \pm 2.31$ | $5.90 \pm 3.66$ | $2.66 \pm 0.76$ | $0.92 \pm 0.05$ | $7.92 \pm 0.02$ | $5.02 \pm 0.93$ |
| | Missing Duplicate | $0.49 \pm 0.04$ | $0.71 \pm 0.04$ | $8.49 \pm 0.27$ | $3.31 \pm 2.16$ | $1.24 \pm 0.35$ | $0.60 \pm 0.02$ | $3.05 \pm 0.01$ | $1.72 \pm 0.30$ |
| | Odds First | $0.55 \pm 0.04$ | $0.99 \pm 0.05$ | $16.63 \pm 1.34$ | $4.49 \pm 2.81$ | $1.67 \pm 0.38$ | $0.80 \pm 0.04$ | $5.48 \pm 0.01$ | $4.57 \pm 1.72$ |
| | Binary Addition | $1.33 \pm 0.05$ | $1.35 \pm 0.04$ | $17.36 \pm 0.45$ | $4.69 \pm 2.94$ | $2.29 \pm 0.75$ | $1.35 \pm 0.05$ | $5.62 \pm 0.01$ | $5.14 \pm 1.48$ |
| | Binary Multiplication× | $1.35 \pm 0.06$ | $1.33 \pm 0.04$ | $17.04 \pm 0.76$ | $4.53 \pm 2.76$ | $2.06 \pm 0.37$ | $1.34 \pm 0.04$ | $5.49 \pm 0.01$ | $4.64 \pm 1.95$ |
| | Compute Sqrt | $0.95 \pm 0.05$ | $1.05 \pm 0.05$ | $12.62 \pm 1.21$ | $3.82 \pm 2.45$ | $1.57 \pm 0.34$ | $0.97 \pm 0.04$ | $4.19 \pm 0.01$ | $3.30 \pm 1.27$ |
| | Bucket Sort†⋆ | $0.58 \pm 0.09$ | $0.98 \pm 0.08$ | $16.75 \pm 2.01$ | $3.54 \pm 2.24$ | $1.34 \pm 0.76$ | $0.79 \pm 0.09$ | $5.64 \pm 0.05$ | $4.52 \pm 1.77$ |

RNN, the hyperparameters we have used are the learning rate, the number of stack symbols, and the random seeds, which gives 15 (tasks) · 2 (number of symbols) · 3 (learning rates) · 10 (seeds) = 900 TPU-units. Moreover, we used an additional 300 TPU-units for the phase transition experiment (Appendix B.2). For the CNN experiment, we used 15 (tasks) · 4 (kernel sizes) · 3 (learning rates) · 10 (seeds) = 1800 TPU-units. For the scaling laws experiment, we used 15 (tasks) · 3 (training steps) · 5 (num layers) · 3 (learning rates) · 10 (seeds) = 6750 TPU-units. Finally, for the autoregresive experiments (excluding the autoregressive Transformer), we used 8 (tasks) · 3 (learning rates) · 10 (seeds) = 240 TPU-units for the RNNs, Stack-RNNs, and LSTMs, i.e., 720 TPU-units, and 8 (tasks) · 3 (computation steps) · 2 (tapes) · 3 (learning rates) · 10 (seeds) = 1440 TPU-units for the Tape-RNNs. Thus, in total we used 20 910 TPU-units. We report all the running times, for each task and architecture, in Table A.2. Note that we aggregated the results over the hyperparameters, which explains the high standard deviation for Transformers (five different positional encodings) and Tape-RNN (three different numbers of computation steps tokens, two different numbers of tapes).

The NDStack-RNN is comparably slow to train. It is roughly four times slower than a classical Stack-RNN, due to the computation of the big tensor $\gamma$, denoting the transition weights between nodes. Furthermore, this tensor takes a lot of space in memory, i.e., the number of values it contains is, once unrolled on the whole trajectory, proportional to $(l + m)^3$, where l and m are the length of the input and output sequences, respectively. Thus, for our training range of 40 and a task where the output length is equal to the input length (e.g., Reverse String (DCF)), the required space is proportional to $(40 + 40)^3 = 512000$. Therefore, with a batch size of 128 and 16 possible actions (i.e., state-symbol pairs), the tensor is roughly of size 33.5Gb using float32 representation, which exceeds the memory size of our GPUs. Consequently, we reduced the batch size to 16 (i.e., 8 times smaller than normal) for this architecture.

# B ADDITIONAL EXPERIMENTS

## B.1 MAIN RESULT

We visualize the performance curves for all tasks in Fig. B.6 and report the means and standard deviations of the test accuracies for all architectures on our tasks in Table B.1. We observe that all the networks, except for Transformers, solve all regular tasks perfectly for all the seeds. However, only a few tasks above beyond regular can be solved (defined by a score larger or equal to 90%) on average, but the hierarchy still persists. For example, the Tape-RNN has much better scores overall, even on average, on the CS tasks compared to the other architectures. However, even though the Tape-RNN can solve more tasks, it still fails on some of them. We hypothesize that it is because the limited set of actions makes the right action trajectory long and difficult to find. Nevertheless, the Tape-RNN is the closest we have to an architecture capable of solving all tasks from regular to CS. Note that networks may be evaluated on unseen sequences even within the training range, as we are sampling from a training distribution and not a finite dataset (as discussed in Section 4). Thus, it is possible

Table B.1: Means and standard deviations (computed over random seeds) of the score (average test accuracy, see Section 4) for the results of the main experiment (see Table 2 and Section 5.1). The † denotes permutation-invariant tasks; the ⋆ denotes counting tasks; the ○ denotes tasks that require a nondeterministic controller; and the × denotes tasks that require superlinear running time in terms of the input length.

| Level | Tasks | RNN | Stack-RNN | Tape-RNN | Transformer | LSTM |
|---|---|---|---|---|---|---|
| R | Even Pairs | **100.0** $\pm$ 0.0 | **100.0** $\pm$ 0.0 | **100.0** $\pm$ 0.0 | 67.7 $\pm$ 15.2 | **100.0** $\pm$ 0.0 |
| | Modular Arithmetic (Simple) | **100.0** $\pm$ 0.0 | **100.0** $\pm$ 0.0 | **100.0** $\pm$ 0.0 | 23.1 $\pm$ 0.8 | **100.0** $\pm$ 0.0 |
| | Parity Check† | 95.5 $\pm$ 14.2 | **100.0** $\pm$ 0.0 | **100.0** $\pm$ 0.0 | 50.4 $\pm$ 0.7 | **100.0** $\pm$ 0.0 |
| | Cycle Navigation† | **100.0** $\pm$ 0.0 | 96.8 $\pm$ 6.6 | **100.0** $\pm$ 0.0 | 33.9 $\pm$ 10.5 | 90.0 $\pm$ 10.8 |
| DCF | Stack Manipulation | 54.5 $\pm$ 1.1 | **93.8** $\pm$ 15.1 | 66.9 $\pm$ 22.6 | 51.1 $\pm$ 8.1 | 58.1 $\pm$ 0.7 |
| | Reverse String | 60.4 $\pm$ 2.3 | **96.2** $\pm$ 12.1 | 90.7 $\pm$ 15.6 | 56.1 $\pm$ 2.7 | 59.5 $\pm$ 1.0 |
| | Modular Arithmetic | 34.6 $\pm$ 4.3 | 75.8 $\pm$ 11.3 | 60.9 $\pm$ 20.7 | 28.2 $\pm$ 3.4 | 53.1 $\pm$ 4.3 |
| | Solve Equation○ | 31.3 $\pm$ 8.7 | 46.1 $\pm$ 7.3 | 37.7 $\pm$ 12.7 | 23.5 $\pm$ 1.6 | 37.1 $\pm$ 14.0 |
| CS | Duplicate String | 50.2 $\pm$ 0.0 | 51.6 $\pm$ 0.6 | **90.0** $\pm$ 16.3 | 52.8 $\pm$ 0.0 | 56.5 $\pm$ 0.9 |
| | Missing Duplicate | 51.7 $\pm$ 0.2 | 53.6 $\pm$ 0.7 | 66.1 $\pm$ 23.3 | 53.8 $\pm$ 1.5 | 53.1 $\pm$ 0.6 |
| | Odds First | 50.4 $\pm$ 0.2 | 51.5 $\pm$ 0.1 | 76.2 $\pm$ 16.6 | 52.7 $\pm$ 0.0 | 54.6 $\pm$ 0.8 |
| | Binary Addition | 49.5 $\pm$ 0.3 | 50.9 $\pm$ 0.9 | 74.3 $\pm$ 10.6 | 51.7 $\pm$ 1.3 | 54.8 $\pm$ 0.4 |
| | Binary Multiplication× | 49.6 $\pm$ 0.2 | 51.4 $\pm$ 0.7 | 54.5 $\pm$ 2.2 | 50.4 $\pm$ 3.6 | 52.9 $\pm$ 0.2 |
| | Compute Sqrt | 53.9 $\pm$ 0.3 | 56.1 $\pm$ 0.3 | 55.6 $\pm$ 1.4 | 51.5 $\pm$ 0.4 | 57.3 $\pm$ 0.2 |
| | Bucket Sort†⋆ | 22.5 $\pm$ 3.6 | 56.6 $\pm$ 16.2 | 33.6 $\pm$ 12.8 | 35.7 $\pm$ 23.1 | 79.4 $\pm$ 21.9 |

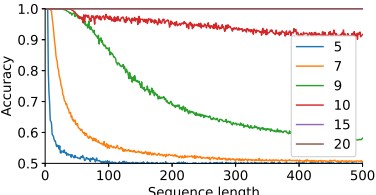
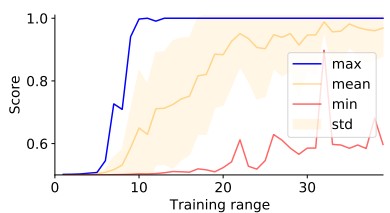

(a) Maximum accuracy (over 10 different random seeds) per length for different training ranges (in color). (b) Score per training range (min, max, and mean taken over 10 random seeds).

Figure B.1: Phase transition effect for the Stack-RNN on `Reverse String` (DCF). For training ranges $N < 10$, the model fails to generalize to longer sequences as it memorizes the training distribution. The score in Fig. B.1b is the accuracy shown in Fig. B.1a averaged over sequence lengths (x-axis).

that some models do not achieve a perfect accuracy within the training range $\mathcal{U}(1, N)$, such as in the `Compute Sqrt` (CS) task (cf. Fig. B.6).

## B.2 PHASE TRANSITION UNDER INCREASING TRAINING RANGES

In this section, we investigate how much training data is needed to learn the data-generating grammars. For certain tasks and architectures we observe a substantial "phase transition" of the test accuracy for different training ranges. For example, Fig. B.1 shows the generalization performance of Stack-RNNs on the `Reverse String` (DCF) task for different training ranges. Clearly, the models fail to learn the data-generating algorithm when only exposed to sequences of length smaller than 10. We hypothesize that for small training ranges the networks overfit to the training data, essentially learning a lookup table to recognize the small finite language. In contrast, larger training ranges encourage the networks to learn the data-generating algorithm that generalizes to sequences of unseen length.

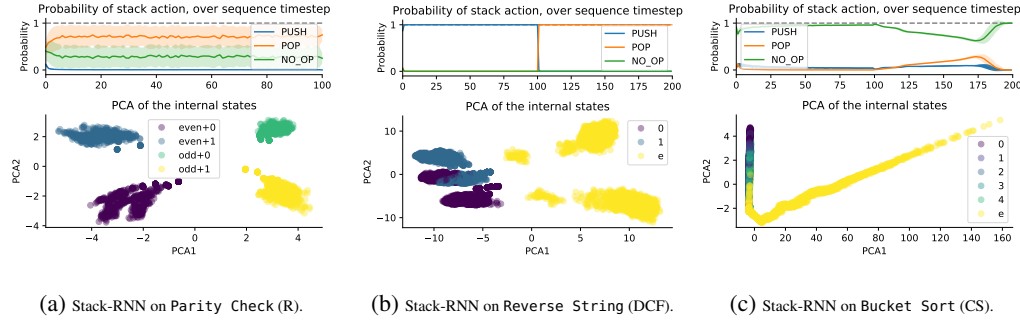

(a) Stack-RNN on `Parity Check` (R).   (b) Stack-RNN on `Reverse String` (DCF).   (c) Stack-RNN on `Bucket Sort` (CS).

Figure B.2: Analysis of the Stack-RNN on the `Parity Check` (R) and `Reverse String` (DCF) tasks. *Left:* The network does not use the stack on `Parity Check` and has the same state structure as a classical RNN. *Right:* The network uses the stack on `Reverse String` and the PCA of the states shows that they are clustered according to the last input token only.

### B.3 ANALYSIS OF STATE- AND MEMORY-DYNAMICS

#### B.3.1 STACK-RNN PROGRAM ANALYSIS

Here, we check that the Stack-RNN, which is our simplest memory augmented network, performs as expected on some of our simple tasks. As mentioned in Section 5.2, Fig. B.2a shows that on the `Parity Check` (R) task the network ignores the stack by only applying NO-OP and POP actions (visualized on an unseen sequence of length 100). We recall that Fig. 3a also showed that the PCA of the states is identical to the one of a simple RNN as it only depends on the last result and the last input token. Moreover, Fig. B.2b demonstrates that on the `Reverse String` task the network always uses the PUSH action until the first empty token is reached, whereupon it starts performing POP actions until the end, which exactly matches the algorithms we would expected a PDA to implement. Note that in this task the number of empty tokens $\varnothing$ we append to the input sequence is equal to the length of the input sequence, and thus the total sequence is of length 200. Finally, Fig. B.2c shows that the Stack-RNN can leverage its probabilistic actions to implement multiple counters and consequently solve (counting) tasks that are beyond deterministic context-free. Concretely, the Stack-RNN uses a mixture of PUSH and NO-OP actions to simultaneously push incremented counters while keeping the previous counter values unchanged. Once the Stack-RNN has consumed the entire input sequence, it switches to a mixture of POP and NO-OP to decrement the counters for the different tokens. Interestingly, once the final token is reached, the Stack-RNN relies mostly on NO-OP actions, since it has learned that the remainder of the output string will contain the token of highest value.

#### B.3.2 TRANSFORMER PROGRAM ANALYSIS

**Setup** The transformer achieves a non-trivial generalization score ($61.9\%$ with baseline $20\%$) on the `Cycle Navigation` (R) task but it does not 'solve' the task (in terms of our definition of score $\geq 90\%$). Therefore, we analyze the network's internals, in particular the attention matrices, to understand which algorithm the network has learned and why it fails on very long sequences. We visualize the attention matrices in Fig. B.3. The color scale is dark blue for $0.0$ and yellow for $0.5$ (not $1.0$ to increase readability). Our transformer networks consist of 5 layers with 8 heads, and we show all of the attention matrices (i.e., $5 \cdot 8$) for a short input sequence of length 8. Note that the actual input to the network is of length 9 since we append an empty token to the input sequence, from which we retrieve the output (see Section 4 and Appendix A).

**Interpretation of the learned algorithm** On the first layer, the model uses only the last empty token and matches it with the tokens 0 and 2 (clearly seen on matrices $(1, 1)$ and $(1, 2)$). Note that the task is to count the number of 0s, the number of 1s, and to do a subtraction modulo 5. This first layer is able to separate the different tokens; the output of the last token for each head is a weighted sum (softmax) of a single value (the token which is attended to), and the weights are all equal to a constant, which means the output of the last token is just the value of the token which is attended to. Then the other tokens have a constant weight over the whole sequence, which means that the output

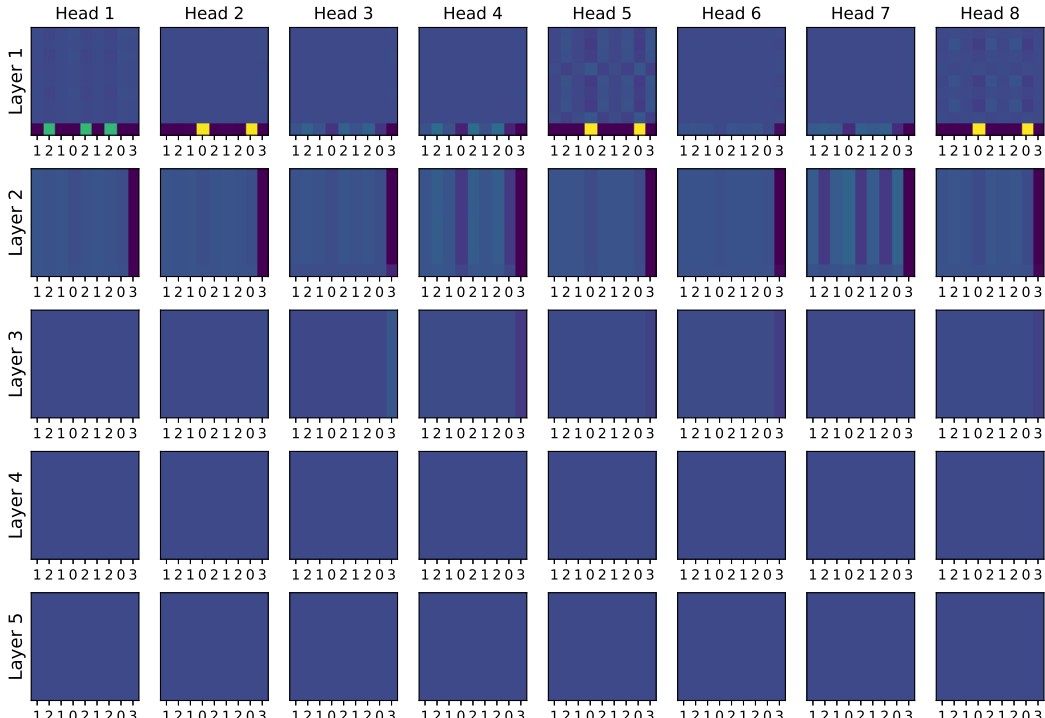

Figure B.3: Analysis of the Transformer attention matrices on the `Cycle Navigation` (R) task, for an input sequence of length 8, used as ticks on the x-axis.

is a weighted sum of all the tokens' values, with the weights also being constant and proportional to the number of occurrence of each token. This will then allow the network to compute the occurrence of each token individually in the linear layer to merge the heads. Fig. B.4 shows a scatter plot of the activations of the network after the first layer for sequences of length 20, which confirms this claim.

**Failure mode for long sequences**   The counting mechanism, which consists in having constant weights over the whole sequence, does not return true counts but frequencies, since they are multiplied by $1/N$ where $N$ is the sequence length (since we are using a softmax). This explains why the network can reasonably generalize to longer lengths, but not perfectly. Looking at Fig. B.4, it means that the structure we see is conserved for longer sequences, but the points become closer and closer, which makes further differentiation into classes more difficult. Fig. B.5 shows the evolution of the activations at the last layer for increasingly long sequences. The classification becomes more challenging because the input activations do not form 5 clear clusters anymore.

### B.4    Convolutional Neural Networks

Bansal et al. (2022) showed that fully convolutional networks achieve near-perfect length generalization on the prefix sum task in the length generalization setting. At the same time, Merrill (2019) theoretically showed that one-layer convolutional networks with maxpool activation functions are incapable of recognizing all regular languages. Therefore, we investigate the length generalization performance of a simple CNN on our benchmark and visualize the maximum and average accuracies over unseen test lengths in Tables B.2 and B.3, respectively. We observe that our simple architecture with kernel sizes in $\{3, 5, 7, 11\}$ (see Appendix A.2) generally fails to solve even regular tasks, achieving the random baseline accuracy in most cases. This confirms the theoretical derivations by Merrill (2019) but contrasts the results presented by Bansal et al. (2022). However, the convolutional architecture that Bansal et al. (2022) consider relies on an adaptive number of layers depending on the sequence length (similar to adaptive computation time (Graves, 2016)) and thus crucially differs from a classical CNN (such as the one we study).

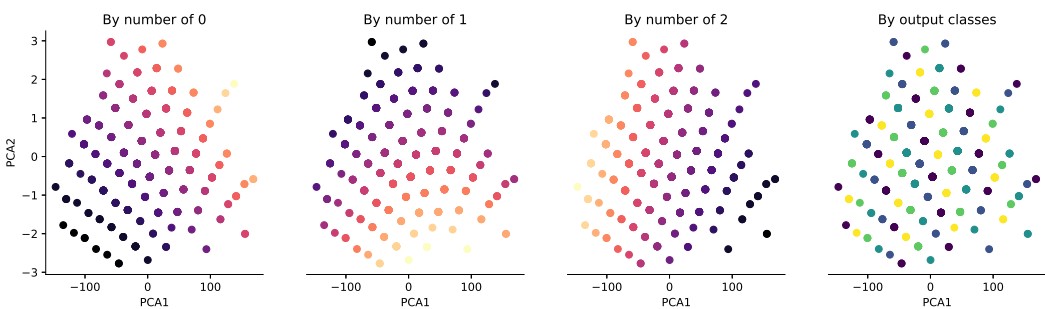

Figure B.4: Analysis of the Transformer activations after the first layer on the `Cycle Navigation` (R) task, for an input sequence of length 8. We report only the first two principal components of 512 sequences. We clearly see that the network learns to count the number of 0s, 1s and 2s and already separates the output classes very well after this first layer.

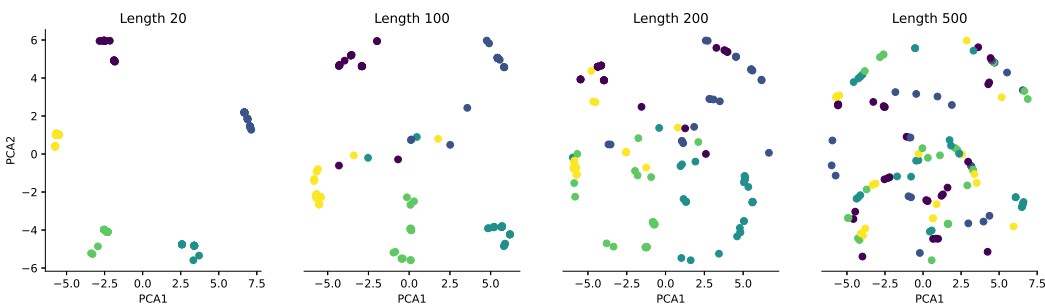

Figure B.5: Analysis of the Transformer activations after the last layer, before the final linear logit layer, on the `Cycle Navigation` (R) task, for an input sequence of length 8. We report only the first two principal components of 512 sequences. We see that the activations form 5 clear clusters for short sequences (length 20), and slowly decay into the center, mixing the clusters together.

## B.5 STACK-LSTMS

We evaluate the experiment suite from Section 5.1 for the Stack-LSTMs and compare their performance with that of Stack-RNNs in Table B.2. We observe that Stack-LSTMs and Stack-RNNs perform roughly equally well when considering the score (maximum accuracy over the 10 initialization seeds), which is not surprising given that both networks are theoretically located on the same level of the Chomsky hierarchy. However, when considering the average Stack-LSTMs perform worse, i.e., our training protocol (gradient descent) is less stable for this architecture, confirming the results of prior work (Hao et al., 2018). We hypothesize that this is due to the fact that LSTMs are more powerful than RNNs (as they can implement k-counter machines) and thus do not always simulate a finite-state automaton. This can be an advantage for more difficult tasks, e.g., counting tasks such as `Bucket Sort` (CS), but also a disadvantage for simpler tasks that only need a finite-state automaton controller to interact with the memory structure. Thus, if the controller is not a finite-state automaton, then its internal state does not interact well with the stack content as it is not part of a finite set, i.e., the interactions are potentially infinite and thus the network does not generalize well beyond the training range.

## B.6 NDSTACK-RNN

We also compared the Stack-RNN and the Stack-LSTM to their non-deterministic counterpart, the NDStack-RNN (which stands for Non Deterministic Stack-RNN) (DuSell & Chiang, 2020; 2022). We observe that this architecture performs well on regular tasks, as the other memory-augmented RNNs. However, it fails on most tasks beyond regular, potentially due to its large action space (24 actions) and the difficulty of training a tensor that tracks all possible stack values. We also don't use unnormalized actions as it requires computing all the probabilities in log-space, and unfortunately JAX (Bradbury et al., 2018) does not support complex tensor multiplications like $einsum$ in log-space. Surprisingly, the architecture solves the `Missing Duplicate` (CS) task perfectly. We tried some ablations and reduced the stack size to only 1: the model still solves the task perfectly. Also, the actions taken by the model on the stack are not binary, leading to mixed computation. We conjecture that the network uses this mixed embedding of values on the stack (reduced to a single cell) efficiently as a memory, on top of the memory of the RNN controller.

## B.7 AUTOREGRESSIVE MODELS

In this paper, we chose to predict the output sequences from empty tokens, but we could also have chosen to predict the sequences autoregressively. Here, we conduct the same set of experiments but in the autoregressive setting to confirm that this aspect of our experimental setup does not impact the architectures' positions in the Chomsky hierarchy. To that end, we use the true output sequence as the input to the model at train time and autoregressive sampling at test time, i.e., we use the predicted output token as the input for the next token until we reach the desired length (more details below for the different architectures). We show the results in Table B.2 with the corresponding means and standard deviations in Table B.3.

### B.7.1 MEMORY-AUGMENTED RECURRENT NEURAL NETWORKS

For the (memory-augmented) RNNs, i.e., RNN, LSTM, Stack-RNN and Tape-RNN, we replace the empty tokens $\varnothing$ with the true output sequence during training. However, to avoid predicting the input, we append an extra empty token between the input sequence (which could contain computation steps) and the true output sequence. We compute the loss with an offset of 1, so the network should predict the first token of the output sequence for the extra empty token, then the second token of the output sequence for the first token, passed now as input, etc. We observe in Table B.2 that these models trained with the auto-regressive system perform very similarly to the ones trained with the empty tokens system (see Table 2 for comparison). That is, this part of the training scheme does not affect the length generalization scores, and therefore does not change where the networks lie on the Chomsky hierarchy.

Table B.2: Score (in percentage, see Section 4), i.e., accuracy averaged over all test lengths and maximized over 10 random seeds (and other hyperparameters, see Appendix A), for the CNN, Stack-LSTM, and NDStack-RNN. The results for the Stack-RNN are identical to Table 2 and are included only for easier comparison with the Stack-LSTM. We consider a model to generalize successfully (bold) if its score $\geq 90\%$. The random accuracy is $50\%$ (except for `Cycle Navigation` (R), `Bucket Sort` (CS), and the two modular arithmetic tasks where it is $20\%$). We denote permutation-invariant tasks with †, counting tasks with ⋆, tasks requiring a nondeterministic controller with ∘, and tasks requiring superlinear running time with ×.

| Level | Tasks | CNN | Stack-RNN | Stack-LSTM | NDStack-RNN |
|-------|-------|-----|-----------|------------|-------------|
| R | Even Pairs | 50.1 | **100.0** | **100.0** | **100.0** |
| | Modular Arithmetic (Simple) | 20.1 | **100.0** | **100.0** | **100.0** |
| | Parity Check† | 50.0 | **100.0** | **100.0** | **100.0** |
| | Cycle Navigation† | 20.1 | **100.0** | **100.0** | **100.0** |
| DCF | Stack Manipulation | 52.4 | **100.0** | **100.0** | 59.2 |
| | Reverse String | 53.0 | **100.0** | **100.0** | 62.4 |
| | Modular Arithmetic | 31.2 | **96.1** | 61.0 | 35.8 |
| | Solve Equation∘ | 20.1 | 56.2 | 67.5 | 47.5 |
| CS | Duplicate String | 50.0 | 52.8 | 59.0 | 51.0 |
| | Missing Duplicate | 51.0 | 55.2 | 55.1 | **100.0** |
| | Odds First | 50.0 | 51.9 | 55.5 | 53.1 |
| | Binary Addition | 49.8 | 52.7 | 56.0 | 53.5 |
| | Binary Multiplication× | 49.9 | 52.7 | 53.3 | 50.9 |
| | Compute Sqrt | 50.2 | 56.5 | 57.6 | 55.0 |
| | Bucket Sort†⋆ | 29.1 | 78.1 | **99.2** | 41.7 |

### B.7.2 TRANSFORMER

In Section 5, we compared the Transformer encoder (used in a non-autoregressive manner) to other sequence prediction models since this allowed for a fair comparison in terms of the experiment setup (all models had to predict the output from empty tokens ∅). However, the original Transformer architecture (Vaswani et al., 2017) also consists of a decoder and processes sequences autoregressively, which is what we evaluate in this section. Similar to memory-augmented RNNs, we observe that the autoregressive model performs roughly as well as the Transformer encoder, except that it fails to solve (or show strong performance) on the permutation-invariant task `Bucket Sort` (CS). This is likely due to the causal masking in the decoder attention, which eliminates permutation-invariance. However, the autoregressive model is capable of solving the `Duplicate String` (CS) perfectly. We hypothesize that this is due to the fact that the relevant input and output tokens are at a fixed length throughout the task and thus the model manages to align the input and output embeddings (using relative positional encodings).

### B.8 SCALING LAWS AND THE CHOMSKY HIERARCHY

Suppose a neural architecture, e.g., a transformer, is limited to a particular algorithmic complexity class (level on the Chomsky hierarchy). In that case, no amount of scaling of the training time, training data, or the number of model parameters can overcome this limitation, thus putting a hard limit on what is possible via scaling. Specific neural architectures might simply be unable to implement certain classes of algorithms in practice (i.e., when considering the training protocol). While we cannot rule out that excessive training, hyper-parameter tuning, or some very delayed grokking effect (Power et al., 2022) would lead to the architecture eventually solving the task, we can show that at least under standard training conditions, we can surface some systematic limitations that are not overcome with more training data or larger models but can be overcome with architectural extensions in line with the theory of computation (i.e., adding a stack or a tape to an RNN).

Table B.3: Means and standard deviations (computed over random seeds) of the score (average test accuracy, see Section 4) for the results of Table B.2. The results for the Stack-RNN are identical to Table B.1 and are included only for easier comparison with the Stack-LSTM. The † denotes permutation-invariant tasks; the ⋆ denotes counting tasks; the ○ denotes tasks that require a nondeterministic controller; and the × denotes tasks that require superlinear running time in terms of the input length.

| Level | Tasks | CNN | Stack-RNN | Stack-LSTM | NDStack-RNN |
|---|---|---|---|---|---|
| R | Even Pairs | $50.1 \pm 0.0$ | $\textbf{100.0} \pm 0.0$ | $\textbf{100.0} \pm 0.0$ | $98.0 \pm 5.7$ |
| | Modular Arithmetic (Simple) | $20.0 \pm 0.1$ | $\textbf{100.0} \pm 0.0$ | $\textbf{100.0} \pm 0.0$ | $78.4 \pm 28.8$ |
| | Parity Check† | $50.0 \pm 0.0$ | $\textbf{100.0} \pm 0.0$ | $\textbf{100.0} \pm 0.0$ | $95.6 \pm 13.6$ |
| | Cycle Navigation† | $20.0 \pm 0.1$ | $\textbf{96.8} \pm 6.6$ | $77.7 \pm 9.5$ | $95.0 \pm 9.1$ |
| DCF | Stack Manipulation | $52.4 \pm 0.0$ | $\textbf{93.8} \pm 15.1$ | $66.2 \pm 17.4$ | $55.1 \pm 2.5$ |
| | Reverse String | $53.0 \pm 0.1$ | $\textbf{96.2} \pm 12.1$ | $\textbf{94.0} \pm 12.4$ | $61.6 \pm 0.6$ |
| | Modular Arithmetic | $30.7 \pm 0.5$ | $75.8 \pm 11.3$ | $51.2 \pm 5.0$ | $28.4 \pm 4.1$ |
| | Solve Equation° | $19.9 \pm 0.1$ | $46.1 \pm 7.3$ | $42.7 \pm 16.0$ | $34.4 \pm 9.7$ |
| CS | Duplicate String | $50.0 \pm 0.0$ | $51.6 \pm 0.6$ | $56.3 \pm 1.3$ | $50.6 \pm 0.2$ |
| | Missing Duplicate | $50.6 \pm 0.1$ | $53.6 \pm 0.7$ | $53.3 \pm 0.7$ | $82.0 \pm 23.3$ |
| | Odds First | $50.0 \pm 0.0$ | $51.5 \pm 0.1$ | $54.4 \pm 0.7$ | $52.2 \pm 0.8$ |
| | Binary Addition | $49.7 \pm 0.2$ | $50.9 \pm 0.9$ | $55.1 \pm 0.6$ | $50.4 \pm 1.3$ |
| | Binary Multiplication× | $49.8 \pm 0.0$ | $51.4 \pm 0.7$ | $52.9 \pm 0.3$ | $50.0 \pm 0.4$ |
| | Compute Sqrt | $50.2 \pm 0.0$ | $56.1 \pm 0.3$ | $57.3 \pm 0.2$ | $54.3 \pm 0.4$ |
| | Bucket Sort†⋆ | $28.4 \pm 0.6$ | $56.6 \pm 16.2$ | $81.7 \pm 19.9$ | $25.9 \pm 7.2$ |

Table B.4: Score (in percentage, see Section 4), i.e., accuracy averaged over all test lengths and maximized over 10 random seeds (and other hyperparameters, see Appendix A), for autoregressive setting. We only include the tasks with output length larger than 1, since the results are identical to those in Table 2 otherwise. We consider a model to generalize successfully (bold) if its score $\geq 90\%$. The random accuracy is $50\%$ (except for Cycle Navigation (R), Bucket Sort (CS), and the two modular arithmetic tasks where it is $20\%$). We denote permutation-invariant tasks with †, counting tasks with ⋆, tasks requiring a nondeterministic controller with ○, and tasks requiring superlinear running time with ×.

| Level | Tasks | RNN | Stack-RNN | Transformer | LSTM |
|---|---|---|---|---|---|
| DCF | Stack Manipulation | 58.0 | **100.0** | 53.2 | 56.9 |
| | Reverse String | 63.1 | **100.0** | 53.5 | 59.4 |
| CS | Duplicate String | 50.3 | 50.7 | **100.0** | 51.7 |
| | Odds First | 50.5 | 51.4 | 54.7 | 53.6 |
| | Binary Addition | 49.8 | 51.7 | 69.0 | 53.9 |
| | Binary Multiplication× | 49.5 | 52.0 | 52.2 | 53.3 |
| | Compute Sqrt | 54.1 | 57.8 | 52.4 | 58.0 |
| | Bucket Sort†⋆ | 28.9 | 52.8 | 40.4 | **92.9** |

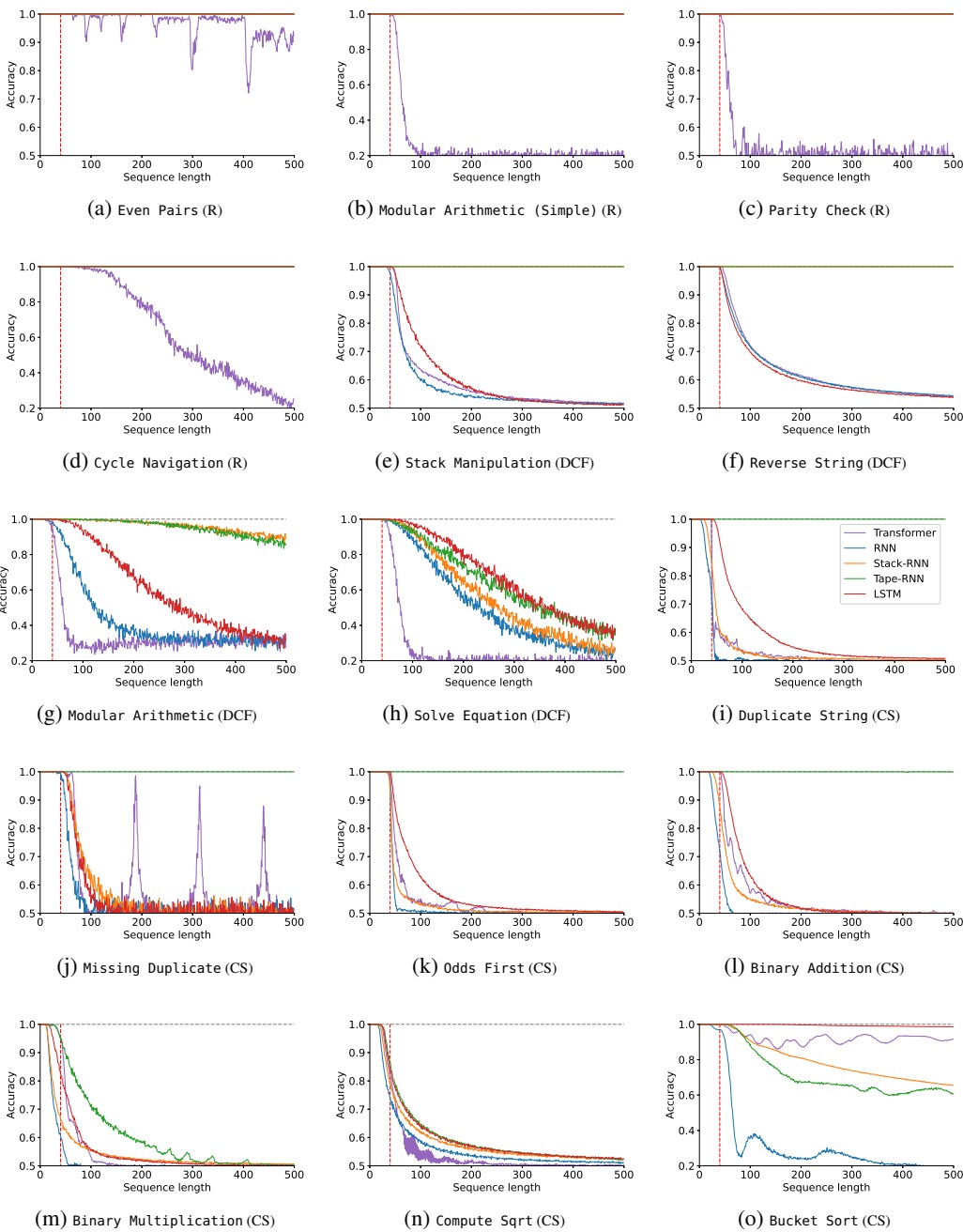

Figure B.6: Performance curves on all tasks. For the Transformer encoder, we pick the best positional encoding for each task (it is considered as a hyperparameter). The dashed vertical red line is the training range, meaning that sequences to the right have not been seen during training and thus measure generalization.

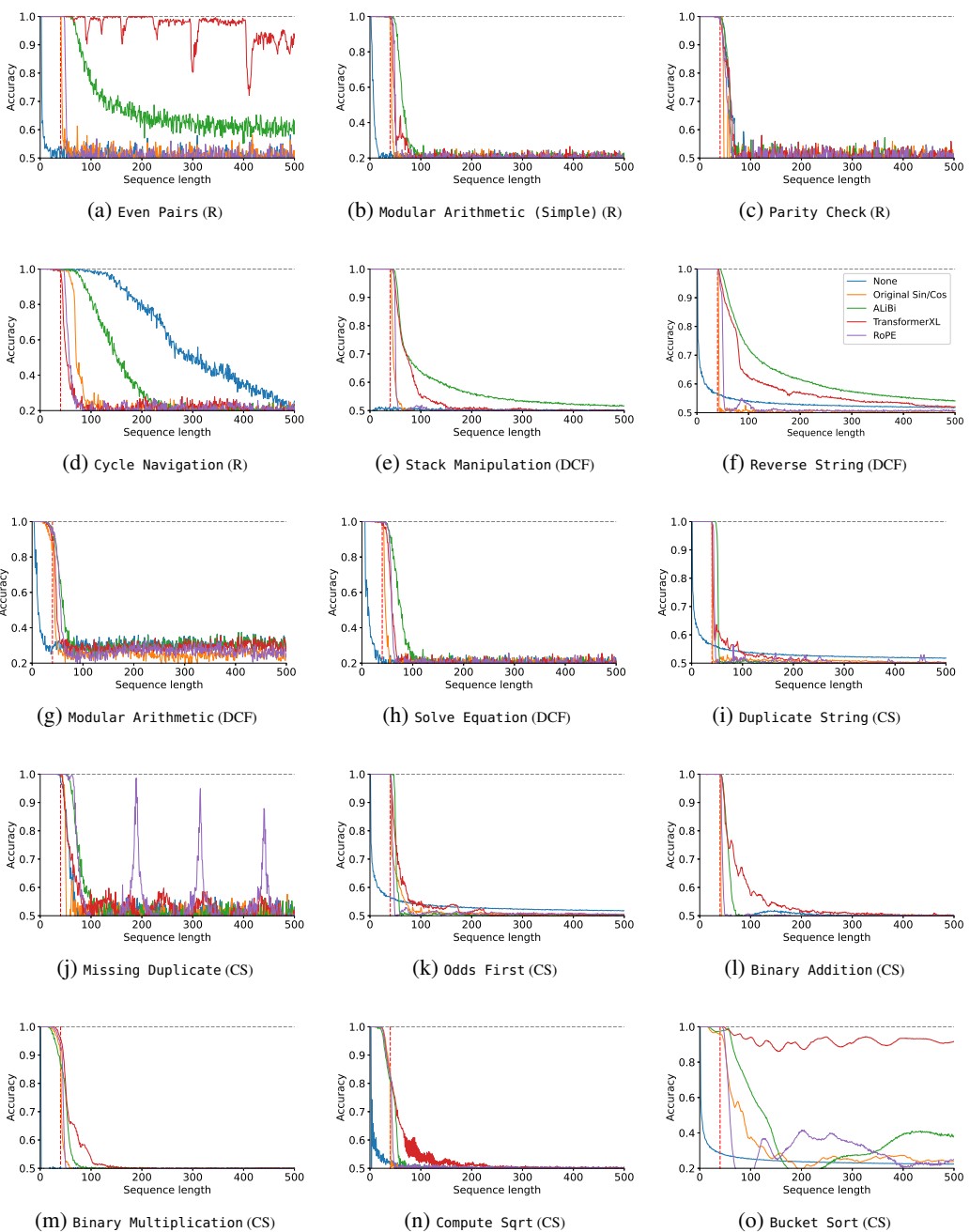

Figure B.7: Performance curves on all tasks for the Transformer encoder architecture, for all the positional encodings we used. The dashed vertical red line is the training range, meaning that sequences to the right have not been seen during training and thus measure generalization.

Table B.5: Means and standard deviations (computed over random seeds) of the score (average test accuracy, see Section 4) for the results in the autoregressive setting (i.e., Table B.4). We only include the tasks with output length larger than 1, since the results are identical to those in Table 2 otherwise. The † denotes permutation-invariant tasks; the ⋆ denotes counting tasks; the ∘ denotes tasks that require a nondeterministic controller; and the × denotes tasks that require superlinear running time in terms of the input length.

| Level | Tasks | RNN | Stack-RNN | Transformer | LSTM |
|-------|-------|-----|-----------|-------------|------|
| DCF | Stack Manipulation | $56.7 \pm 0.2$ | $\mathbf{100.0} \pm 0.00$ | $51.7 \pm 1.3$ | $55.8 \pm 0.1$ |
| | Reverse String | $62.4 \pm 0.1$ | $\mathbf{99.3} \pm 0.2$ | $52.3 \pm 1.6$ | $58.1 \pm 0.1$ |
| CS | Duplicate String | $50.2 \pm 0.1$ | $50.5 \pm 0.1$ | $\mathbf{92.7} \pm 4.7$ | $51.4 \pm 0.1$ |
| | Odds First | $50.4 \pm 0.1$ | $51.0 \pm 0.02$ | $52.3 \pm 1.4$ | $52.9 \pm 0.4$ |
| | Binary Addition | $49.4 \pm 0.2$ | $50.5 \pm 0.5$ | $60.2 \pm 4.2$ | $52.9 \pm 0.6$ |
| | Binary Multiplication$^\times$ | $47.5 \pm 0.5$ | $50.5 \pm 1.3$ | $50.8 \pm 1.3$ | $52.5 \pm 0.5$ |
| | Compute Sqrt | $53.7 \pm 0.3$ | $57.5 \pm 0.3$ | $54.5 \pm 0.5$ | $57.7 \pm 0.5$ |
| | Bucket Sort$^{\dagger\star}$ | $21.8 \pm 3.4$ | $35.6 \pm 8.7$ | $32.1 \pm 4.3$ | $\mathbf{93.0} \pm 6.7$ |

To provide a more detailed empirical result, we train the transformer encoder with relative positional encodings for increasing numbers of in-distribution data points (100 000, 1 000 000, and 10 000 000 training steps) and increasing numbers of layers (5, 10, 15, 20, 25). Table B.6 shows that the score, i.e., the accuracy averaged over all (unseen) test lengths and maximized over 10 seeds and 3 learning rates, generally does not increase with more training data or more parameters. Thus, in this empirical setup, the Chomsky hierarchy implies a hard limit on scaling laws (Kaplan et al., 2020).

Table B.6: Score (see the caption of Table 2) for the Transformer with relative positional encodings and different numbers of layers and training steps (i.e., training data). We observe that the Chomsky hierarchy implies a hard limit on scaling laws (Kaplan et al., 2020), i.e., more in-distribution data and/or more parameters do not lead to better length generalization performance.

| Level | Task | Training Steps | Layers | | | | |
|---|---|---|---|---|---|---|---|
| | | | 5 | 10 | 15 | 20 | 25 |
| R | Even Pairs | 100 000 | **90.2** | 84.7 | 74.8 | 50.1 | 50.1 |
| | | 1 000 000 | **90.0** | 78.6 | 78.7 | 50.1 | 50.1 |
| | | 10 000 000 | **97.5** | 89.7 | 89.5 | 50.1 | 50.1 |
| | Modular Arithmetic (Simple) | 100 000 | 21.7 | 21.6 | 21.5 | 20.5 | 19.9 |
| | | 1 000 000 | 21.2 | 21.6 | 22.6 | 22.1 | 20.0 |
| | | 10 000 000 | 21.3 | 22.1 | 24.8 | 24.4 | 20.0 |
| | Parity Check† | 100 000 | 50.3 | 50.4 | 50.1 | 50.1 | 49.9 |
| | | 1 000 000 | 51.5 | 50.3 | 50.1 | 50.1 | 50.1 |
| | | 10 000 000 | 52.0 | 50.5 | 50.1 | 50.1 | 50.1 |
| | Cycle Navigation† | 100 000 | 24.5 | 25.4 | 23.7 | 22.9 | 23.2 |
| | | 1 000 000 | 22.9 | 22.5 | 23.6 | 22.0 | 23.7 |
| | | 10 000 000 | 22.8 | 22.9 | 25.9 | 23.2 | 23.3 |
| DCF | Stack Manipulation | 100 000 | 54.9 | 55.4 | 55.3 | 54.5 | 54.9 |
| | | 1 000 000 | 54.0 | 54.4 | 53.7 | 54.9 | 56.2 |
| | | 10 000 000 | 52.8 | 51.9 | 53.0 | 52.9 | 49.6 |
| | Reverse String | 100 000 | 61.6 | 61.2 | 58.0 | 56.1 | 56.8 |
| | | 1 000 000 | 59.8 | 60.6 | 61.0 | 60.1 | 58.8 |
| | | 10 000 000 | 56.5 | 58.7 | 57.2 | 52.1 | 50.0 |
| | Modular Arithmetic | 100 000 | 31.4 | 31.7 | 31.9 | 31.5 | 31.6 |
| | | 1 000 000 | 28.9 | 27.9 | 31.0 | 31.0 | 31.0 |
| | | 10 000 000 | 27.5 | 30.8 | 31.0 | 31.0 | 31.0 |
| | Solve Equation° | 100 000 | 22.4 | 21.3 | 21.5 | 22.2 | 19.9 |
| | | 1 000 000 | 23.0 | 22.4 | 21.6 | 21.8 | 19.9 |
| | | 10 000 000 | 22.8 | 21.6 | 21.9 | 23.7 | 20.1 |
| CS | Duplicate String | 100 000 | 52.5 | 53.6 | 53.1 | 55.3 | 53.6 |
| | | 1 000 000 | 51.9 | 51.5 | 53.1 | 53.7 | 54.2 |
| | | 10 000 000 | 51.9 | 51.1 | 51.7 | 51.3 | 50.0 |
| | Missing Duplicate | 100 000 | 53.9 | 52.7 | 52.6 | 52.1 | 50.0 |
| | | 1 000 000 | 53.2 | 53.0 | 50.0 | 50.0 | 50.0 |
| | | 10 000 000 | 52.1 | 52.5 | 50.0 | 50.0 | 50.0 |
| | Odds First | 100 000 | 52.5 | 53.4 | 52.9 | 52.2 | 53.2 |
| | | 1 000 000 | 52.5 | 53.4 | 52.9 | 52.7 | 53.5 |
| | | 10 000 000 | 51.8 | 52.8 | 52.0 | 50.0 | 50.0 |
| | Binary Addition | 100 000 | 52.3 | 54.7 | 51.7 | 51.2 | 50.8 |
| | | 1 000 000 | 53.3 | 55.9 | 54.8 | 54.8 | 53.8 |
| | | 10 000 000 | 52.9 | 52.9 | 51.8 | 54.4 | 51.1 |
| | Binary Multiplication× | 100 000 | 52.3 | 52.3 | 51.9 | 51.2 | 51.0 |
| | | 1 000 000 | 52.5 | 53.1 | 53.4 | 53.4 | 54.0 |
| | | 10 000 000 | 51.8 | 53.0 | 52.0 | 52.8 | 50.6 |
| | Compute Sqrt | 100 000 | 52.8 | 52.7 | 52.6 | 52.0 | 52.7 |
| | | 1 000 000 | 52.3 | 53.4 | 53.7 | 54.0 | 53.5 |
| | | 10 000 000 | 51.6 | 52.7 | 53.4 | 53.3 | 53.0 |
| | Bucket Sort†★ | 100 000 | **90.9** | **93.9** | **93.0** | **91.4** | 87.5 |
| | | 1 000 000 | **93.1** | **91.6** | **92.1** | **93.7** | **93.0** |
| | | 10 000 000 | 84.3 | 81.9 | 80.4 | 89.5 | 65.1 |

