# OpenReview forum: "Neural Networks and the Chomsky Hierarchy"
_ICLR.cc/2023/Conference — ICLR 2023 notable top 25%_

### Official Review · Reviewer_vMMY · 2022-10-13

**Confidence:** 2
**Correctness:** 3
**Technical Novelty And Significance:** 3
**Empirical Novelty And Significance:** 3
**Recommendation:** 6

**Clarity, Quality, Novelty And Reproducibility:**

The paper is clear and easy to read.

The empirical evaluation quality is high.

The novelty is to relate the Chomsky hierarchy and neural network out-of-distribution generalization and to conduct an extensive empirical study.

The paper has details for experiments, and it open-sources the benchmark.

**Strength And Weaknesses:**

***Strength:***

The best strength is the scale of the evaluation.
It includes a variety of models and representative tasks in the hierarchy.

The Chomsky hierarchy is essential in computation theory. So it is reasonable to find its relation to the neural network to help understand the out-of-distribution generalization for neural networks.

***Weakness:***

I have comments and questions.

(1) This paper uses the "limit" of generalization to group neural network architectures.
It indicates that even if a model architecture works on a representative task for a Chomsky hierarchy level, it might not work on other tasks on the level.
Though it is mentioned in the "limitation," it might be worth attention because it differs from the classic automatons' case.

(2) The introduction mentions that the length generalization problem "subsumes all computable problems."
Is it for classic models like automatons, or is it also correct for neural networks?

For example, suppose a neural network is trained with short inputs starting with 'a' and can generalize to long inputs beginning with 'a'. However, it is still unclear whether it can generalize to short inputs starting with 'b.'


**Summary Of The Paper:**

This paper conducts a large-scale extensive empirical study to practically investigate whether insights in the computation theory can predict the out-of-distribution generalization limits for neural networks.

It uses more than 10 thousand models and 15 tasks to evaluate the performance of program induction neural network models on sequence prediction tasks defined in the Chomsky hierarchy. The models include state-of-the-art architectures, such as RNN, LSTM, Transformer, and memory-augmented networks, such as Stack-RNN and Tape-RNN.

It finds the relation between models and hierarchy on out-of-distribution generalization. For example, memory-augmented networks can solve high-hierarchy tasks.

It also finds that using more training data does not enable the generalization higher up in the hierarchy for some architectures.

It opensources a novel benchmark for length generalization, designed to accurately pinpoint the architectures' problems.


**Summary Of The Review:**

This paper has an extensive empirical study with high coverage of models in interest and tasks in the Chomsky hierarchy.

It has significant contributions and strengths, as mentioned above.

---

> ### Author Response · Authors · 2022-11-17
> **Response**
>
> Thank you for your thoughtful comments.
>
>
> **Does the length generalization problem also subsume all computable problems for neural networks or only for classical automata?**
>
> Yes, as we mention in the introduction `[Generalization of sequence prediction] is of particular importance since it subsumes all computable problems`. Note that this claim only concerns the problem formulation and is independent of the concrete model used to solve the problem (i.e., an automaton or a neural network). Thus, whether a network that generalizes from short to long inputs for sequences starting with $a$ also generalizes for inputs starting with $b$ cannot be answered in general but only with respect to a particular task.

---

### Official Review · Reviewer_6CTT · 2022-10-25

**Confidence:** 4
**Correctness:** 3
**Technical Novelty And Significance:** 4
**Empirical Novelty And Significance:** 4
**Recommendation:** 8

**Clarity, Quality, Novelty And Reproducibility:**

Clarity
---------
There is a lack of clarity on the Chomsky's hierarchy of transduction tasks vs Chomsky's hierarchy of language tasks, but beyond this I found the paper to be clear and an enjoyable read. The Algorithm A.1 was especially beneficial in understanding the work and would suggest adding to the main paper if space permits.

Quality
---------
I am quite happy with the quality of this work.

Novelty
---------
I find this to be novel work and I believe the field would greatly benefit from this work especially in understanding the limits of our architectures and training mechanisms.

Reproducibility
-------------------
They have open sourced a length-generalization benchmark. Yes, it should reproducible.

**Strength And Weaknesses:**

Strengths
-------------

Formal languages are relatively under-explored in AI literature, so it is quite encouraging to see efforts in this direction. This work does a great job of identifying Chomsky’s hierarchy as a possible proxy for studying inductive biases in neural architectures and their corresponding training mechanisms for sequence prediction tasks.

This work also marries well with existing literature on the Turing completeness of neural architectures. The empirical evidence from this work, covers an important gap that was missing from the theoretical literature.

Although the results are slightly underwhelming (some architectures cannot solve tasks at their supposed level), it nevertheless demonstrates the limits of these neural architectures for sequence prediction tasks.

The results from this work have also demonstrated some interesting insight on memory augmented neural architectures, which are often less favoured for sequence prediction tasks.

Weaknesses
-----------------

The main weaknesses of this work is the disambiguation of transduction and recognition. I can appreciate that the authors have made an effort to disambiguate between the two concepts which in itself is a non-trivial feat but I don’t think it’s evident from the introduction and abstract that sequence prediction would translate to Finite State Transduction (FST) Machines instead of Finite State Automata (FSA) Machines.

I would suggest dedicating an entire section before the background describing transduction vs recognition and FST/FSA. This was partially addressed in the background section but it is tucked away at the end and requires a couple reads to finally get the greater picture. However this doesn’t take away from the paper, as the motivation to reduce the problem to a transduction problem is justified.

- The generative grammar section could be better explained and would suggest discussing more about Chomsky’s rules as they are also vital in categorizing across Chomsky’s hierarchy. I also suggest placing the generative grammar section before the formal language section or combining the two, as this might read better.
- The per sequence accuracy equation in page 5, doesn’t seem to make sense. I would suggest revising this. Same applies for the equation in Algorithm A.1.
- The stack manipulation example in Table A.1 seems incorrect. Are you pushing an empty string? If so, would that be the result?
- The claim regarding the hard limits on scaling laws is not well explored in this paper and is only mentioned in the introduction and conclusion. I would suggest elaborating on this in the results. I think this is sadly one of the shortfalls of this paper.

**Summary Of The Paper:**

The paper empirically groups sequence to sequence neural architectures models according to the Chomsky Hierarchy. They achieve this by evaluating 10250 neural architectures models across 15 transduction tasks spanning the entire Chomsky Hierarchy.

They demonstrate that memory augmented architectures tend to fall higher in the Chomsky Hierarchy than those without structures memory (e.g. stack or tape).

**Summary Of The Review:**

I find this work to be an important contribution towards understanding the limits of our neural architectures and their corresponding training mechanisms for sequence prediction tasks. The results on memory-augmented neural architectures were similarly insightful. All in all, I am happy to accept this paper.

I am willing to reconsider my recommendation to an oral presentation if:
- Clarity around transduction and recognition is improved
- The per sequence accuracy equation in page 5 is revised
- The generative grammar section formally defines grammar and includes some dummy examples of Chomsky grammar rules.
- The claim regarding the hard limits on scaling laws is well justified.

---

> ### Author Response · Authors · 2022-11-17
> **Response**
>
> Thank you for your positive review and constructive feedback!
>
> **Can you provide more disambiguation between transduction and recognition and their relation to sequence prediction?**
>
> Yes, thank you for the suggestion. We have updated our manuscript to formally introduce finite-state automata and finite transducers in the background section. Moreover, we clarified that a (deterministic) transducer induces a (deterministic) mapping between two formal languages, which corresponds to the function we want our models to learn in our tasks. The relationship between language recognition and transduction has been well-studied theoretically. Thus, to keep our manuscript clear and easy to follow, we do not introduce this distinction in the abstract and introduction but introduce it as an essential detail in the background section.
>
>
> **Can you provide a better explanation of the generative grammar section?**
>
> Yes, we clarified the generative grammar section by formally defining production rules and describing how the Chomsky hierarchy arises from categorizing the allowed relationships between terminals and non-terminals in the grammar. Moreover, we provide a concrete example of the production rules for the (binary) palindrome language and explaine how it relates to the Chomsky hierarchy by analyzing the characteristics of its production rules.
>
> Thank you for helping us clarify this important aspect of our paper!
>
>
> **Is the per-sequence accuracy equation on page 5 correct?**
>
> Yes, we believe that the accuracy equation is correct. We consider one-hot encoded sequences $y_{ij}$ and models $p_\theta(\cdot | x)$ as conditional distributions over the next possible sequence continuations. For token $i$, we compute the predicted label as $argmax_j p_\theta(\cdot | x_j)$ and the corresponding target from the one-hot encoded sequence as $argmax_j y_{ij}$. Then we check if the predicted label is equal to the target with the indicator function. Finally, we average over all tokens in the output sequence by summing over $i$ and dividing by the length of the sequence $|y|$.
>
>
> **Can you elaborate on your claim regarding the hard limits of scaling laws?**
>
> Yes, suppose a neural architecture, e.g., a transformer, is limited to a particular algorithmic complexity class (level on the Chomsky hierarchy). In that case, no amount of scaling of the training time, training data, or the number of model parameters can overcome this limitation, thus putting a *hard limit* on what is possible via scaling. Specific neural architectures might simply be unable to implement certain classes of algorithms in practice (i.e., when considering the training protocol). Our current results do not allow us to make a strong claim of this kind, but through our study, we identified some tasks for which more than simple scaling is needed. While we cannot rule out with absolute certainty that excessive training, hyper-parameter tuning, or some very delayed grokking effect (Power et al., 2022) would lead to the architecture eventually solving the task, we can show that at least under standard (extensive) training conditions, we can surface some systematic limitations that are not overcome with more training data or larger models but can be overcome with architectural extensions in line with the theory of computation (i.e., adding a stack or a tape to an RNN).
>
> To provide a more detailed empirical result, we train the transformer encoder with relative positional encodings for increasing numbers of in-distribution data points ($100000$, $1000000$, and $10000000$ training steps) and increasing numbers of layers (5, 10, 15, 20, 25). Table B.4 shows that the score, i.e., the accuracy averaged over all (unseen) test lengths and maximized over 10 seeds and 3 learning rates, generally does not increase with more training data or more parameters. Thus, in this empirical setup, the Chomsky hierarchy implies a hard limit on scaling laws (Kaplan et al., 2020).
>
>
> **Is the stack manipulation example in Table A.1 correct?**
>
> Sorry, no, we did not update the table with the correct version provided in the task description of Section A.3. Thank you for your careful study of our paper! The example in the table is now correct.

---

### Official Review · Reviewer_unzx · 2022-10-25

**Confidence:** 5
**Correctness:** 3
**Technical Novelty And Significance:** 3
**Empirical Novelty And Significance:** 3
**Recommendation:** 8

**Clarity, Quality, Novelty And Reproducibility:**

**Clarity and Quality:** The paper is written very well. The introduction and background is written beautifully in a way that's succinct while discussing crucial subtleties (such as the distinction between transduction vs. recognition). The experiments and analyses are rigorous and well executed.

**Originality:** While there exist works in the literature that also tackle the problem of length generalization (some listed above), the angle that this paper takes on it (angle of theory of computation) is novel and worthy of study.

**Reproducibility:** To the extent I can tell, the paper provides enough details to reproduce the findings. Perhaps some low-level details on the types of initializations is missing, but the code release remedies this.








**Strength And Weaknesses:**

**STRENGTHS**
* **Important and timely study:** Investigating the generalization patterns of sequence models is a very important research direction and comes at a time when we need more of this type of work that bridges theory and practice, and when safety concerns (that might be caused by lack of robustness under distribution shifts) are growing.
* **Crisp problem description and task design:** The authors are very systematic in the way they pick the tasks and define length generalization as the locus of their analysis. There are many ways of framing this problem, and the way the authors have done it seems very sensible.
* **Relatively comprehensive experiments:** While the experiments and ablations aren't exhaustive (some of the important components that are omitted will be described below), the experiments are still quite comprehensive and has a reasonable focus (i.e. on tools to augment sequence models with tools such as stacks and tapes).
* **Nontrivial and interesting insights:** The reported limitations of existing architectures have important consequences on the forecasting of AI progress. For example, it looks like naive scaling (i.e. increase model size and in-distribution data size) alone won't solve length generalization unless we get clever with training details.
* **Nice interpretability analysis:** The interpretability analyses of RNNs with tapes and stacks confirm the intuitions one might have about why these models are able to display length generalization.

**WEAKNESSES**
* **Missing components of ablation:** I'd like to highlight some components that seem to be missing in the experiments. (I'd like to note that no study is fully comprehensive -- I'm listing these in order to provide a balanced view of the weaknesses of the submission)
  * Convolutional architectures missing: Bansal et al. [3] have shown that fully convolutional networks show near-perfect length generalization on the prefix sum task (akin to parity). It looks like convolutional networks are also relevant in a study that focuses on the effect of architecture on length generalization.
  * Effect of pretraining: [4] shows that pretraining data could have an effect on the inductive bias displayed by transformer models. Likewise, [2] shows that large language models (when prompted properly) can display nontrivial length generalization, which is largely due to the pretraining procedure instead of scale.
  * Effect of depth: While RNN-like architectures "reason" through timesteps, transformers "reason" along their depth. Hence, it makes sense that a 5 -transformer -block architecture won't have the capacity to handle large depths unless there's a shallow shortcut (like there is in Bucket sort). It'd be interesting to see either very deep transformers, or universal transformers [5] investigated in this work too.
  * No focus on autoregressive models: Autoregressive generation is what enables scratchpad strategies in transformers, which could help them attain length generalization capabilities.
* **Missing references:** I'd like to highlight two prior work that appear to be very relevant to the current submission:
  * "Unveiling Transformers with LEGO: a synthetic reasoning task" This paper proposes a task that allows for testing encoder-only transformers' ability to display length generalization. The paper has interesting insights into what goes wrong when transformers fail at length generalization and which components (like pretraning) seem to help.
  * "Exploring length generalization in large language models" This paper explores length generalization too, and finds that vanilla transformers lack this ability even at an extremely large scale. They show, however, that in-context learning and the use of a scratchpad significantly improves transformers' ability to posses length generalization.

**QUESTIONS TO AUTHORS**
* What exactly is the rightmost column in Table 1?
* The square root computation doesn't appear to be marked as requiring superlinear time in Table 3. Could you explain what linear algorithm solves this task?




[1] Zhang, Yi, et al. "Unveiling Transformers with LEGO: a synthetic reasoning task." arXiv preprint arXiv:2206.04301 (2022).

[2] Anil, Cem, et al. "Exploring length generalization in large language models." arXiv preprint arXiv:2207.04901 (2022).

[3] Bansal et al. End-to-end Algorithm Synthesis with Recurrent Networks: Logical Extrapolation Without Overthinking

[4] Wu, Yuhuai, et al. "Lime: Learning inductive bias for primitives of mathematical reasoning." International Conference on Machine Learning. PMLR, 2021.

[5] Dehghani, Mostafa, et al. "Universal transformers." arXiv preprint arXiv:1807.03819 (2018).


**Summary Of The Paper:**

**High level motivation:** The authors are interested in understanding what it takes for machine learning systems to reliable generalize to novel circumstances, especially when those are out-of-distribution.

**Research question:** Do insights from the theory of computation (specifically the Chomsky Hierarchy) inform us about the limits of neural network generalization in practice?

**Operationalization:** The authors have:
* **(tasks)** Designed 15 tasks that cover regular, context-free, context-sensitive and recursively enumerable languages (i.e. require automatons with computational requirements that are associated with the aforementioned grammar types to solve),
* **(success definition)** Picked length generalization (ability to learn from short problem instances to generalize to longer ones ) as the target capability,
* **(architecture)** Run extensive experiments on RNNs (augmented with a stack and a tape), LSTMs and transformers (in total 10250 models),
* **(insights)** Demonstrated that the computational capabilities (i.e. access to stack or memory, a state etc.) of models roughly predict which tasks they display length generalization on.

**Contributions:**
* An extensive empirical study spanning multiple architectures and tasks,
* The release of a dataset to evaluate length generalization on,
* The depiction of serious failure modes regarding length generalization that can partially be explained by the (mis)match between a given model's computational capabilities and the language class the task at hand belongs to.
* Evidence that some architectural modifications (like the addition of external memory modules) might mitigate length generalization deficiencies.



**Summary Of The Review:**

The authors run an extensive empirical study checking whether theory of computation can be used to predict which architectures will generalize on different tasks. To a large extent, there's a match between a given architecture's computational limitations and the requirements (in the theory of computation sense) a task requires.

That this is a well-executed paper on an important and timely topic. I believe ICLR would be a better conference with this paper.

---

> ### Author Response · Authors · 2022-11-17
> **Response (1/2)**
>
> Thank you for your detailed review and positive feedback!
>
> **What is the effect of depth on the generalization capability for transformers?**
>
> We investigated the relation of the Chomsky hierarchy and scaling laws, i.e., increasing the in-distribution training data and the number of parameters (concretely, the number of layers), and summarized the results in Table B.4. Concretely, we considered transformer encoders with relative positional encodings for increasing numbers of data points ($100000$, $1000000$, and $10000000$ training steps) and increasing numbers of layers (5, 10, 15, 20, 25). Table B.4 shows that the score, i.e., the accuracy averaged over all (unseen) test lengths and maximized over 10 seeds and 3 learning rates, generally does not increase with more training data or deeper networks.
>
>
> **What is the effect of pretraining and prompting for transformers?**
>
> An (independent) follow-up paper by a different group (citation omitted since it cites the pre-print of  our work and would thus break anonymity) has investigated the effect of pretraining and prompting of transformers with respect to the Chomsky hierarchy using our length generalization benchmark. The paper finds that pretrained large language models (LLMs) are less aligned with the Chomsky hierarchy than memory-augmented RNNs in the sense that they can solve some complex tasks (e.g., binary addition) but not simpler ones (e.g., modular arithmetic). Moreover, the paper finds that both larger models and longer prompts generally lead to higher accuracy. However, the paper also shows that even extensive pretrained transformers (up to 176 billion parameters) are unable to solve most of the length generalization tasks in our benchmark.
>
> Therefore, given that an entire paper has been written on this particular aspect of our experimental study, we believe that including such an investigation is out of scope. We will, of course, discuss the paper in the camera-ready version.
>
>
> **How do convolutional architectures perform on your length generalization benchmark?**
>
> Bansal et al. (2022) showed that fully convolutional networks achieve near-perfect length generalization on the prefix sum task in the length generalization setting. At the same time, Merill (2019) theoretically showed that one-layer convolutional networks with maxpool activation functions are incapable of recognizing all regular languages. Therefore, we investigate the length generalization performance of a simple CNN (5 convolutional layers of 32 channels each and a final MLP layer) on our benchmark and visualize the maximum and average accuracies over unseen test lengths in Tables B2. and B.4, respectively. We observe that our simple architecture with kernel sizes in $\{3, 5, 7, 11\}$ (see Appendix A.2 for the full architecture) generally fails to solve even regular tasks, achieving the random baseline accuracy in most cases. This confirms the theoretical derivations by Merrill (2019) but contrasts with the results presented by Bansal et al. (2022). However, the convolutional architecture that Bansal et al. (2022) consider relies on an adaptive number of layers depending on the sequence length (similar to adaptive computation time Graves (2016)) and thus crucially differs from a classical CNN (such as the one we study) and is thus beyond the scope of our work.
>
>
> **How do autoregressive models perform on your length generalization benchmark?**
>
> We considered the non-autoregressive setting since it provides the most straightforward way of assessing the length generalization capabilities of neural architectures. However, we agree that the autoregressive setting is relevant for length generalization, which is why we had already conducted experiments with the autoregressive transformer. We now also investigated the performance of (memory-augmented) RNNs in Appendix B.7 (Tables B.4 and B.5). The results confirm that the experimental setting (i.e., autoregressive vs. non-autoregressive) does not affect the length generalization scores of the models, meaning that it also does not change their location in the Chomsky hierarchy. Note that the Tape-RNN results are still missing since the experiment is still running. However, the interim results indicate no fundamental change in scores, and we will update the manuscript with the results as soon as the experiments have been completed.
>
>
> **What linear algorithm solves square root computation?**
>
> Newton's method computes the integer square root with a quadratic convergence rate (see https://en.wikipedia.org/wiki/Integer_square_root#Algorithm_using_Newton's_method), which corresponds to a time complexity of $\mathcal{O}(\log(\log(n)))$ to obtain the required precision of $\log(\sqrt{n})$ digits, which means it is logarithmic in the number of digits.

---

> > ### Comment · Reviewer_unzx · 2022-11-26
> > **Thank you for your response**
> >
> > I thank the authors for their detailed response!
> >
> > The answers reaffirm my perception of the paper, which I believe is a strong submission and would clearly benefit the NeurIPS community if accepted. Hence, I maintain my score of 8.

---

> ### Author Response · Authors · 2022-11-17
> **Response (2/2)**
>
> **You should reference two relevant prior works [1, 2].**
>
> Thank you for pointing out these two related concurrent papers. We added a short discussion of these papers, which investigate the length generalization capabilities of pretrained large language models on synthetic reasoning tasks, to the related work section.
>
>
> **What is the rightmost column in Table 1?**
>
> The rightmost column corresponds to the example languages provided in the *Formal languages* paragraph in Section 3. We made this more explicit in the table caption.

---

### Official Review · Reviewer_U2az · 2022-10-25

**Confidence:** 4
**Correctness:** 3
**Technical Novelty And Significance:** 3
**Empirical Novelty And Significance:** 3
**Recommendation:** 8

**Clarity, Quality, Novelty And Reproducibility:**

Clarity: The description of the transduction task is a little unclear. Other than that, the paper is well-written.

Quality: The empirical results cover a wide range of models and tasks.

Novelty: Although I think some pieces of the conclusions have been proposed in prior works (which tend to be less systematic and mostly case-by-case), I view the current paper under review as a systematic empirical survey, with extensive independent verifications. (I think this is still valuable information, though in the response section to this review, the authors can feel free to clarify exactly what results are new in this paper, versus what results are already known in prior works.)

Reproducibility: In the supplementary materials, the authors included codes. I assume that the authors will publicly release the codes.

**Strength And Weaknesses:**

Strengths:

1. Some interpretation results (in Section 5.2) on some combinations of tasks and trained models are strong evidences that the models did learn some groundtruth algorithms to solve the tasks.

2. The paper reports the empirical average accuracy of models trained and evaluated on a wide range of tasks.


Weaknesses:

The set of interpreted/interpretable combinations of tasks and models are limited compared with the set of all claims made in this paper. In particular, when a model achieves higher-than-random but much-lower-than-perfect accuracy on some task, it remains unclear whether the model implemented a correct algorithm (up to small numerical errors), or it implemented a crude approximation. In particular, such approximation may reside in a lower level of the Chomsky hierarchy (e.g. approximating a context-sensitive task with some context-free level algorithm). The authors' claim is somewhat ambiguous on this point (see my 2nd question below).


Questions for the authors:

1. In Figure 2(b), for the modular arithmetic task, why is there a slight increasing trend for transformers when the sequence length is between ~150 and ~300? Does this phenomenon robustly persist if you re-run it? (In comparison, all other curves in Figure 2 show a general decreasing trend.)

2. In Section 5.4, 2nd paragraph, the author stated that “Table 3 shows that Transformers generally succeed on permutation-invariant tasks: They solve Bucket Sort (CS) and demonstrate non-trivial generalization on Cycle Navigation (R).” However, Table 3 shows that on Cycle Navigation, the average accuracy is just 61.9% (random baseline = 20%). This is indeed higher than the baseline, but still seems too low to convincingly justify that the model correctly solved the task. (See my comment in the Weaknesses section above.) Could the authors provide more evidence on whether transformers indeed learns some correct algorithm for Cycle Navigation, or propose a revision of the claim that "Transformers generally succeed on permutation-invariant tasks" to make the statement more precise?

3. In Appendix A.1, the authors discusses the limitations of predicting the set of possible next tokens by mentioning that it is “incompatible with the (standard sequence) prediction setting and thus of lesser interest to the wider machine learning community.” Could you share your thoughts on whether we *should* pay more attention to the type of next token prediction tasks whose target is a set rather than a single token? In particular, I am curious to learn:
- In principle, is it fundamentally different from current approaches? (i.e. is "training with multiple labels for each sample" a good approximation of "training with a single label, different each time, for multiple steps"?)
- Practically, I can imagine some applications like auto-completing sentences in email apps or text editors. When it provides multiple candidates, it seems to me that this can be done with models trained either with single-target, or with multi-target?



**Summary Of The Paper:**

This work conducts extensive experiments, showing that the ability of common neural network architectures to learn formal languages can roughly be characterized by the Chomsky hierarchy.

Another contribution of this paper is that, the results empirically confirmed that augmenting neural network architectures with auxiliary memory (e.g. stack or tape) is helpful for generalization in the formal language setting. (The paper under review is not the first paper that proposes this idea, but it is still good that the paper provides more evidence on this with extensive empirical results.)


**Summary Of The Review:**

In summary, this work reports extensive empirical results. This work also includes a small number of case studies that shows the models did learn some groundtruth algorithms to solve the tasks in a limited number of settings. I think these strengths justify at least a weak accept. Satisfactorily resolving the above-mentioned weaknesses and questions may potentially justify a higher rating.



Edit after reading authors' response:

I think the authors sufficiently addressed my main concerns about correctness and clarity by updating their paper with some additional in-depth analyses. So I increased the overall rating from 6 to 8.

---

> ### Author Response · Authors · 2022-11-17
> **Response (1/2)**
>
> Thank you for your positive feedback and interesting questions!
>
>
> **Please clarify your claim that “transformers generally succeed on permutation-invariant tasks”.**
>
> Thank you for pointing this out. Transformers achieve an accuracy of 61.9% on the *Cycle Navigation* task, which, according to our criterion, does not count as ‘solving the task’. What we meant to say is that transformers are generally more successful on permutation-invariant tasks than on tasks that require positional information. Therefore, we have reformulated the claim to "Transformers are most successful on permutation-invariant tasks".
>
> Moreover, we have analyzed the attention matrices of the transformer without positional on the *Cycle Navigation* task and explained this reasonable but not perfect length generalization. The network learns a counting mechanism by using constant weights on the whole input sequence, which returns a weighted sum equal to the sum of each token's frequency (count/sequence length) multiplied by their value. This trick works for frequencies that have been seen during training and can extend to slightly different frequencies (exactly as neural networks generalize to embeddings close to those seen during training) but not to too precise ones, which occur for large sequences in which the frequencies are much more numerous and therefore much more precise. The model has trouble differentiating them and slowly fails to generalize as the sequence length increases.
>
>
> **What happens when models achieve higher-than-random but much-lower-than-perfect accuracy on a certain task?**
>
> Our study aims to provide an empirical upper bound on model capabilities, i.e., showing which task classes are out of reach for particular architectures. For models that achieve higher-than-random but much-lower-than-perfect accuracy on a task, we cannot entirely rule out that very extensive hyper-parameter tuning would lead to 'solving the task', implying that we would are seeing the effects of sub-optimal hyper-parameters or initialization, numerical instabilities in the model, etc. In the discussion section, we have stated this as an explicit limitation that needs to be considered when interpreting our results.
>
> Models in such less-than-perfect cases might have learned a partial or incomplete version of a correct algorithm, an algorithm that only generalizes well for a subset of the data, or simply a numerically unstable version of a correct algorithm. Teasing these cases apart can be very labor-intensive and challenging (often, entire publications are written on analyzing a single model on a single task). Thus, while it could be interesting in the future to see if there is some systematic issue in some less-than-perfect models and relate that to certain aspects of the training protocol, such an analysis is beyond the scope of our current paper, and we abstain from speculating at this point.
>
> The one exception to this in the paper is our analysis of transformers' failure of generalization (Section 5.4), which we have prioritized because we think it is very timely and important, and we were able to formulate a clear hypothesis regarding the origin of the observed failure mode.
>
>
> **Can you clarify the description of the transduction tasks?**
>
> Yes, thank you for the suggestion. We have updated our manuscript to formally introduce finite-state automata and finite transducers in the background section. Moreover, we clarified that a (deterministic) transducer induces a (deterministic) mapping between two formal languages, which corresponds to the function we want our models to learn in our tasks. The relationship between language recognition and transduction has been well-studied theoretically. Thus, to keep our manuscript clear and easy to follow, we do not introduce this distinction in the abstract and introduction but introduce it as an essential detail in the background section.

---

> > ### Comment · Reviewer_U2az · 2022-11-19
> > **Thank you for the response**
> >
> > I think the authors sufficiently addressed my main concerns about correctness and clarity by updating their paper with some additional in-depth analyses. So I increased the overall rating from 6 to 8.

---

> ### Author Response · Authors · 2022-11-17
> **Response (2/2)**
>
> **Can you clarify the difference between the novel results in this paper and those of prior work?**
>
> Yes, prior work investigated the length generalization capabilities of specific architectures, both theoretically and empirically. However, in contrast to previous theoretical work, our focus is on whether and to which degree models are capable of achieving their theoretical capabilities (which do not consider the training protocol as an important practical factor that limits a model’s effective complexity class, often significantly more than what a theoretical analysis of the model only would suggest). The main challenge is systematically picking a battery of tasks that allows for a structured and meaningful investigation of abstract model capabilities (their practically realizable complexity class) while evaluating concrete tasks only. We use the theory of computation and the Chomsky hierarchy to guide our task design to achieve this. This allows us, and is the main novelty, to go beyond a simple collection of tasks (some of which are solvable, some of which are not) to a *systematic characterization*. Some of our results on particular tasks and with particular architectures have been reported before in the literature, e.g., that transformers fail to solve *Parity Check* (Hahn, 2020). However, our main contribution is the systematic and comprehensive comparison of network performance and behavior across tasks, i.e., transformers generally fail on tasks that are not permutation-invariant; Stack-RNNs can solve many context-free tasks, and the use of a stack significantly extends a model’s effective complexity class compared to a plain RNN (which, in theory, should be able to learn to emulate a stack internally and thus lie in the same theoretical complexity class). Moreover, we introduced the Tape-RNN and showed that it could solve tasks up to the context-sensitive level.
>
>
> **Why is there a slight increasing trend for transformers when the sequence length is between ~150 and ~300 (Figure 2b)?**
>
> This is most likely an artifact of the stochastic nature of our training protocol (e.g., numerical reasons, weight initialization, etc.). Figure B.4 (g) provides a more granular view of the transformer's performance on the *Modular Arithmetic* task, showing that the increase is part of a larger (smooth) fluctuation around 25% accuracy. Indeed, most trajectories across our different hyperparameters and random seeds (not shown as we only visualize the best model) monotonically decrease over increasing sequence lengths.
>
>
> **Should we pay more attention to next-token prediction tasks whose target is a set rather than a single token?**
>
> We chose the standard setup of predicting a single token instead of a set of tokens so that our results would be relevant to the broader machine learning community. Nevertheless, as you point out, studying the task of predicting a set of possible next tokens is interesting in its own right, both from a research and an application perspective. To answer your questions:
> * **Are the two settings fundamentally different?** The multi-target setting can be approximated in various ways, e.g., by sampling a token from the set of possible tokens at every step as you suggested or by computing the loss over the top-k logits (similar to top-k accuracy in computer vision). However, the two approaches are not entirely equivalent, and thus the results will not be immediately comparable across different settings.
> * **Are there practical applications that could benefit from multi-target models?** While it is true that a multi-target model would more naturally lend to applications that require multiple candidates (e.g., auto-completion), such behavior can also be implemented by repeatedly sampling from a single-output model with the same input. Which of the two approaches works better will depend on the exact experiment setting.
>
>
> **Will you release the code?**
>
> Yes, we release an open-source implementation of our models, tasks, and training and evaluation suite.

---

### Official Review · Reviewer_VcFp · 2022-10-25

**Confidence:** 4
**Correctness:** 3
**Technical Novelty And Significance:** 2
**Empirical Novelty And Significance:** 3
**Recommendation:** 6

**Clarity, Quality, Novelty And Reproducibility:**

The paper is clear and likely to be reproducible. The novelty is somewhat limited due to its empirical nature and most of the properties highlighted are well-known by now.

**Strength And Weaknesses:**

Strength
======
- A nice empirical study with clear designs and messages sent across.
- Results are useful, pointing to directions of memory, which has been somewhat neglected in the past 8 years.

Weaknesses
==========
- As with any empirical study, there is never enough experiments to 100% confirm the points. The authors have clearly articulated that in the limitations section.
- The authors pointed out that learning procedure (e.g., gradient-based) would have an important impact on the generalizability, but this was not analyzed further.
- If the empirical nature is the main focus, some statistical analysis would be useful.
- It would have been much more insightful if some in depth analysis of why models behave they way they do (e.g., LSTM after all is a RNN with state transition dynamic, Transformer is similar to feedforward net, but with clever design of attention, we can make it behave like it has some memory), and why some theoretical properties do not hold in the experiments (e.g., attention is Turing complete, but why Transformer doesn't seem to work well in designed tasks?).

**Summary Of The Paper:**

The paper provides empirical evidences of generalizability of modern neural architectures over synthetic tasks following the Chomsky hierarchy. The results generally agree with theoretical results, highlighting (again) the need for external memory.

**Summary Of The Review:**

An useful empirical study to highlight the need for memory to clime the Chomsky hierarchy. More in depth analysis of model behaviors and learning dynamics would be much more beneficial to the community, which is currently sticking around Transformer for empirical successes.

---

> ### Author Response · Authors · 2022-11-17
> **Response**
>
> Thank you for your helpful assessment of our work.
>
>
> **How does the learning procedure affect the generalizability?**
>
> Understanding this question is precisely the goal of our work. Prior studies have investigated the theoretical capabilities of neural architectures (showing, e.g., that RNNs are Turing complete) but without considering the entire training protocol (i.e., gradient-based learning, parameter initialization, etc.). Our paper bridges this gap by conducting a large-scale empirical study of the length generalization capabilities of neural architectures in practice, which accounts for the entire training protocol.
>
>
> **Can you please provide in-depth analyses of why models behave the way they do?**
>
> Yes, we have already provided in-depth analyses of model behavior in Sections 5.2, 5.4, and B.3. As mentioned above, our paper completes theoretical analyses of neural architectures, which often make unrealistic assumptions, such as infinite precision or infinite computation time. Thus, while it is theoretically correct that attention is Turing complete, our paper empirically demonstrates that transformers can fail to recognize even regular languages. Moreover, we discussed that this is (partially) linked to the fact that the positional encodings are out-of-distribution for longer sequence lengths in Section 5.4.

---

### Author Response · Authors · 2022-11-17
**General Response**

We thank the reviewers for their detailed and insightful feedback, which has helped us to improve many aspects of our submission.

We are particularly pleased that the reviewers consider our paper novel and important (`R-unzx`, `R-6CTT`), our empirical study well-designed (`R-VcFp`, `R-unzx`) and extensive (`R-U2az`, `R-unzx`, `R-vMMY`), our results useful (`R-VcFp`) and insightful (`R-6CTT`), our paper clear (`R-VcFp`, `R-vMMY`) and well-written (`R-U2az`, `R-unzx`), and that `the field would greatly benefit from this work` (`R-6CTT`) and `ICLR would be a better conference with this paper` (`R-unzx`).

According to the reviewers’ suggestions, we added the following experiments (training an additional $10710$ models) to the manuscript:
* We investigated the relationship between scaling laws (i.e., depth and in-distribution data scaling) and the Chomsky hierarchy (Section B.8) and showed that increasing numbers of parameters (depth) and/or training data does not enable transformers to solve more tasks.
* We evaluated performance of convolutional neural networks on our length generalization benchmark (Section B.4) and showed that they fail to recognize regular languages, as theoretically demonstrated by Merrill (2019).
* We studied the length generalization of the autoregressive analogues of all our models (Section B.7) and showed that this training and evaluation scheme did not change the length generalization scores of the models and, therefore, their position in the Chomsky hierarchy.
* We analyzed the algorithm learned by transformers on the *Cycle Navigation* task (Section B.3.2) and showed that the transformer learns the “correct” algorithm but fails to generalize to longer sequences due to the softmax function in the attention.

Moreover, we clarified the following sections:
* We formally characterized the relationship between formal language recognition and transduction and precisely described the difference between the two with respect to our setting (Section 3).
* For completeness of the background section, we added the relationship between the levels of the Chomsky hierarchy and the corresponding generative grammar class to Table 1 (Section 3).
* We clarified the claim about the performance of transformers on permutation-invariant tasks, stating that they are more successful on permutation-invariant tasks but cannot solve them in general (Section 5.4).
* We clarified the caption of Table 1 to explain the right-most column (Section 3) by referencing the corresponding paragraph of the paper.
* We fixed the stack manipulation example in Table A.1 by reversing the input stack to represent the bottom-to-top order.
* We added a discussion of two concurrent related papers on length generalization for pretrained large language models (Section 2).

We are looking forward to clarifying any further questions the reviewers may have.

---

> ### Comment · Reviewer_U2az · 2022-11-19
> **Thanks and followup suggestions**
>
> Thank you for these updates!
>
> Suggestion for improving one of these newly added sections (Section B.3.2) in your next revision of the paper:
>
> I think this analysis is helpful for understanding the empirical performance.
> However, Figure B.4 remains quite unclear to me:
> - What do the colors on the dots represent, and if they are counts or class labels, which color means what number?
> - While the caption states that "we report only the first two principal components of 512 sequences", I am fairly sure that in each sub-plot, there are far less than 512 dots. Do many of the 512 sequences have almost overlapping first two principle components of their activations?
> - The caption states that "we clearly see that the network [...] already separates the output classes very well after this first layer." However, what I see in the rightmost sub-plot is that, the activations corresponding to all these classes are still mostly mixed together. Could you clarify those?

---

### Decision · Program_Chairs · 2023-01-20

**Decision:**

Accept: notable-top-25%

**Justification For Why Not Higher Score:**

As with any probing study, the conclusions are always "somewhat provisional" --- e.g. it's difficult to exclude whether tweaks to the hyperparameters or architecture could not change the results (especially so for the negative results in the paper). Various reviewers pointed out weaknesses with both the experimental scaffolding and architectural ablations (and the authors have been forthright for some of these as well).

**Justification For Why Not Lower Score:**

The paper is a good large-scale, thorough study of the empirical performance of various versions of transformer-like architectures. These kinds of studies are likely to be of interest to a large fraction of the ICLR community.

**Metareview: Summary, Strengths And Weaknesses:**

The paper proposed a fairly thorough experimental scaffolding, exploring the capabilities of transformer-based architectures to learn (and generalize) on various tasks "sorted by difficulty" through the Chomsky hierarchy. The authors explored both vanilla transformer-based architectures, and various "memory-augmented" variants, and compile a thorough set of both positive and negative results. The reviewers, throughout the course of the discussion, suggested various additional experiments, and expository clarifications, which substantially improved the paper. The consensus is this is a good paper, that will be of interest to many in the conference.

**Note From Pc:**

if the above contains the word "oral" or "spotlight" please see: "oral" presentation means -> notable-top-5% and "spotlight" means -> notable-top-25%. As stated in our emails, we are disassociating presentation type from AC recommendations

**Summary Of Ac-Reviewer Meeting:**

N/A